# Mutual inhibition between PTEN and PIP3 generates bistability for polarity in motile cells

Satomi Matsuoka [1,2,4] & Masahiro Ueda [1,2,3,4]

Phosphatidylinositol 3,4,5-trisphosphate (PIP3) and PIP3 phosphatase (PTEN) are enriched mutually exclusively on the anterior and posterior membranes of eukaryotic motile cells. However, the mechanism that causes this spatial separation between the two molecules is unknown. Here we develop a method to manipulate PIP3 levels in living cells and used it to show PIP3 suppresses the membrane localization of PTEN. Single-molecule measurements of membrane-association and -dissociation kinetics and of lateral diffusion reveal that PIP3 suppresses the PTEN binding site required for stable PTEN membrane binding. Mutual inhibition between PIP3 and PTEN provides a mechanistic basis for bistability that creates a PIP3-enriched/PTEN-excluded state and a PTEN-enriched/PIP3-excluded state underlying the strict spatial separation between PIP3 and PTEN. The PTEN binding site also mediates the suppression of PTEN membrane localization in chemotactic signaling. These results illustrate that the PIP3-PTEN bistable system underlies a cell's decision-making for directional movement irrespective of the environment.

[1] Laboratory for Cell Signaling Dynamics, RIKEN QBiC, 6-2-3, Furuedai, Suita, Osaka 565-0874, Japan. [2] Laboratory of Single Molecule Biology, Graduate School of Frontier Biosciences, Osaka University, 1-3 Yamadaoka, Suita, Osaka 565-0871, Japan. [3] Laboratory of Single Molecule Biology, Graduate School of Science, Osaka University, 1-1 Machikaneyama, Toyonaka, Osaka 560-0043, Japan. [4] Present address: Laboratory for Cell Signaling Dynamics, RIKEN BDR, 6-2-3, Furuedai, Suita, Osaka 565-0874, Japan. Correspondence and requests for materials should be addressed to S.M.(email: s.matsuoka@riken.jp)

**D**ynamic anterior–posterior polarity is a hallmark of eukaryotic motile cells. The signaling system responsible for the polarity is largely shared among a wide spectrum of eukaryotes, ranging from mammalian immune cells to social amoebae *Dictyostelium discoideum*[1–3]. A common key signaling molecule is phosphatidylinositol 3,4,5-trisphosphate (PIP3). PIP3 is generated via the phosphorylation of phosphatidylinositol 4,5-bisphosphate (PIP2) by phosphoinositide-3-kinase (PI3K) and degenerated via its dephosphorylation into PIP2 by phosphatase and tensin homolog deleted from chromosome 10 (PTEN) on the cell membrane[4–6]. PIP3 is enriched at the prospective anterior side, where it serves as a binding site for Pleckstrin homology (PH) domain-containing proteins such as PKB (also known as Akt) and other proteins, including Rac activator DOCK2, which promote pseudopodium formation[7–11]. In general, the spatiotemporal dynamics of PIP3 is an essential determinant of eukaryotic motile behavior.

A remarkable characteristic of the PIP3 dynamics is the clear confinement of the enriched region, known as the PIP3 patch or PIP3-enriched domain[12,13]. Such an all-or-none distribution is beneficial for the confined activation of pseudopodium formation, and thus effective directed migration. This confinement is lost in the absence of PTEN in *Dictyostelium discoideum* cells, which fail to suppress the lateral pseudopod or make directional movement[5,14]. PTEN is localized exclusive of the PIP3-enriched domain in an area known as the PTEN-enriched domain. The PIP3-enriched and PTEN-enriched domains are separated by a clear border where PIP3 and PTEN levels change abruptly[15–17]. It has been proposed that the steep enrichment is gained by amplification through a positive-feedback loop[18–20]. PIP3 enhances the activity of Ras through pseudopod formation, which recruits PI3K, which contains a Ras-binding domain to further produce PIP3[21,22]. F-actin is not a prerequisite for this amplification[15]. On the other hand, PTEN produces PIP2 on the cell membrane to further recruit PTEN, which contains a PIP2-binding motif[23–25]. Although these two positive-feedback loops require coupling with each other to avoid merging of the PIP3-enriched and PTEN-enriched domains, interactions between the anterior and posterior signaling molecules have hardly been taken into account.

One interaction that could explain the clear separation is mutual inhibition of the anterior and posterior signaling molecules. Previous studies have predicted that PTEN membrane localization is negatively regulated by PIP3 by using a mathematical model that describes self-organized traveling waves of the PIP3-enriched and PTEN-enriched domains[15,19]. Such negative regulation, together with the lipid phosphatase activity of PTEN, leads to a mutually inhibitory relationship between PTEN and PIP3. The mutual inhibition between the two positive-feedback loops can provide a mechanistic basis for bistability, a feature of systems that show ultrasensitive switching between two metastable states where the chosen positive-feedback loop is exclusively activated[26,27]. However, there is no compelling evidence or mechanistic explanation for the negative regulation of PTEN by PIP3. Moreover, it is counterintuitive that the substrate causes the exclusion of the enzyme from the substrate-enriched region. In addition, PTEN membrane localization can be suppressed without PIP3 in *pi3k*-null *D. discoideum* cells in response to a chemoattractant, 3′,5′-cyclic adenosine monophosphate (cAMP)[28]. Therefore, a mechanistic issue to be addressed is how the membrane localization of PTEN is regulated, especially in relation to the local PIP3 level as well as the chemoattractant stimulation.

In this study, we aim to clarify the causality between PIP3 and PTEN levels on the cell membrane. By combining the genetic and pharmacological manipulation of PI3K activity and simultaneous live-cell imaging of the spatiotemporal dynamics of PIP3 and PTEN, we give evidence for the negative regulation of PTEN membrane localization by PIP3. Replacement of *D. discoideum* PTEN with a *Homo sapiens* homolog defective in the negative regulation demonstrate that mutual inhibition leads to clear spatial separation between PIP3 and PTEN. Single-molecule imaging reveals that the negative regulation is mediated by a specific binding site for PTEN that is inactivated not only by PIP3 but also by cAMP stimulation. These results illustrate that PTEN works as a component of the bistable system to generate a digitized signal of the confined PIP3 enrichment and thereby determine a cell's motile behavior irrespective of the environment.

## Results

**Clear spatial separation between PTEN and PIP3.** Anterior–posterior polarization in *D. discoideum* requires PTEN (DdPTEN), the loss of which causes constitutive PIP3 enrichment on the whole-cell membrane and multiple pseudopods projecting simultaneously toward all directions, thus compromising cell motility[5,14] (Fig. 1a; Supplementary Fig. 1a, b). HaloTag-tagged DdPTEN labeled with tetramethylrhodamine (TMR) was localized on the cell membrane, but excluded from the leading edge of migrating cells, as reported previously[5]. The exclusion occurred even in the presence of a comparable amount of PIP2, as visualized with green fluorescent protein (GFP)-tagged Nodulin (GFP-Nodulin), to the lateral and posterior membranes (Fig. 1b). Instead, the exclusion was always accompanied with an enrichment of PIP3, as visualized with eGFP-tagged Pleckstrin-homology domain of PKB (PH_PKB-eGFP), making a clear spatial separation between PTEN and PIP3.

We analyzed the spatial distribution of PIP3 in mutant cell lines that have dysregulated enzymatic activity or PTEN membrane localization by replacing endogenous DdPTEN with HaloTag-tagged variants that elevated the basal levels of PIP3 (Fig. 1c, d). The expression of DdPTEN-Halo complemented the *pten*-null defects, such that the polarity and motility were almost the same as those of wild type (Fig. 1c, e, f, left; Supplementary Table 1). The PIP3-enriched domain at the leading edge was confined and separated from the PTEN-enriched domain, ensuring pseudopod projection almost exclusively at the anterior side. On the other hand, the expression of one DdPTEN variant, DdPTEN_G129E, which lacks lipid phosphatase activity but shows intact membrane localization, could not recover the *pten*-null defects[23,25] (Fig. 1c, e, f, middle). The PIP3 level was increased uniformly on the cell membrane, where DdPTEN_G129E was co-localized, allowing pseudopod projections at the lateral and posterior sides. The expression of another variant, *Homo sapiens* PTEN (HsPTEN), which has an intact catalytic site but shows lower membrane-binding affinity than DdPTEN, also recovered the polarity and motility to a certain extent[24] (Fig. 1c, e, f, right). These cells exhibited an expanded and gradually changing PIP3 distribution without obvious membrane localization of HsPTEN, resulting in a significant increase in lateral and posterior pseudopods, which terminated the cell migration temporarily. The defects were more severe with regards to chemotaxis (Supplementary Movie 1; Supplementary Movie 2; Supplementary Movie 3; Supplementary Fig. 2; Supplementary Table 1). These results suggest that confined PIP3 enrichment requires the appropriate membrane localization of PTEN along with lipid phosphatase activity.

We confirmed these observations were independent of the morphological anterior–posterior polarity by treating the cells with latrunculin A, an actin polymerization inhibitor. In the cells expressing DdPTEN, PIP3-enriched domains arose occasionally on the cell membrane without any directional preferences (Fig. 1g–i, left), and DdPTEN exhibited mutually exclusive

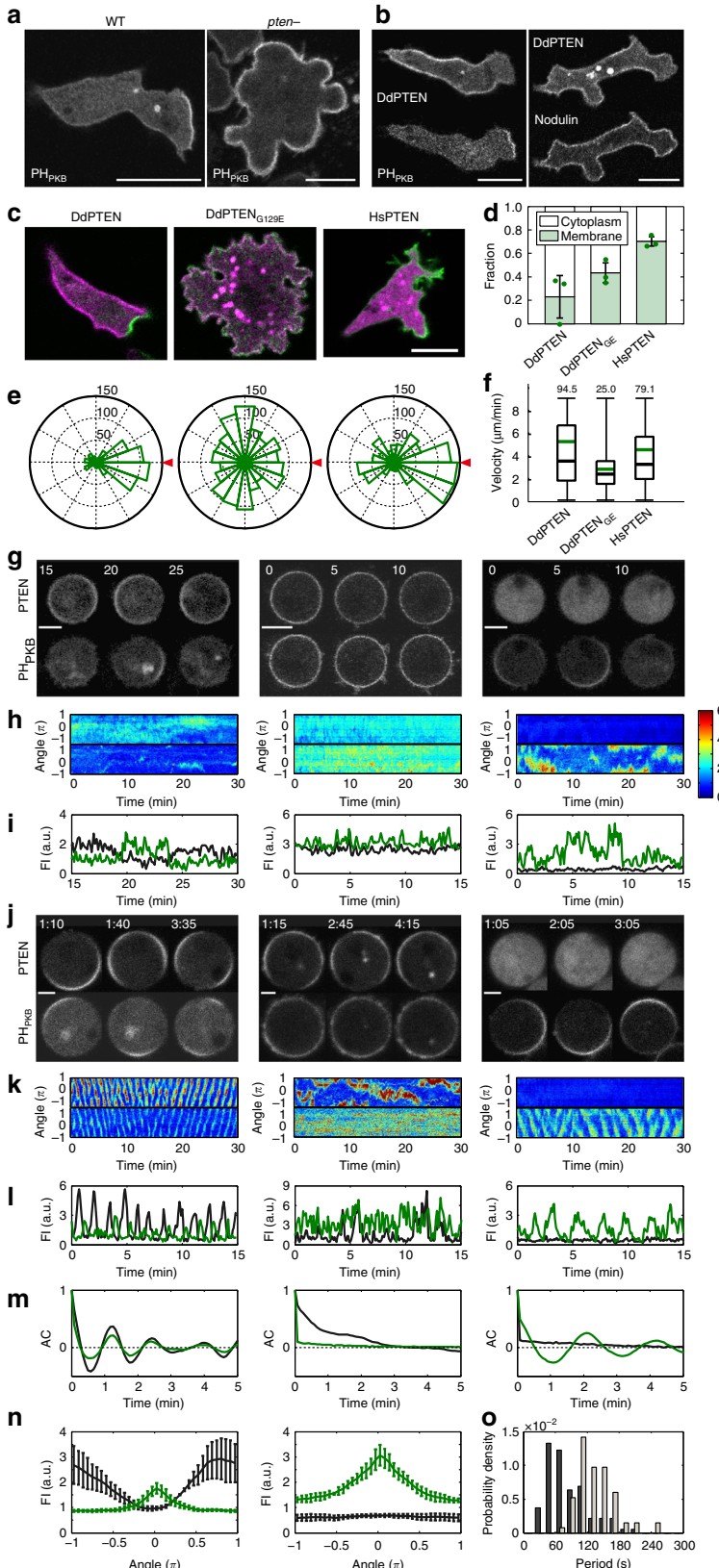

membrane localization with PIP3. In the cells expressing DdPTEN$_{G129E}$, the PIP3 level was constantly and uniformly increased on the whole-cell membrane (Fig. 1g–i, middle), and DdPTEN$_{G129E}$ exhibited co-localization with PIP3. In the cells expressing HsPTEN, the PIP3-enriched domain arose in a

broadened and prolonged manner (Fig. 1g–i, right), and HsPTEN remained in the cytoplasm without any detectable membrane localization irrespective of PIP3 levels. For the purpose of statistically comparing the spatiotemporal characteristics of the PIP3-enriched domains, all cells were further treated with 4 mM

**Fig. 1** Membrane localization of PTEN is critical for the confinement of PIP3 enrichment and cell motility. **a** Confocal images of living wild-type (left) and *pten*-null (right) cells expressing $PH_{PKB}$-eGFP. **b** Confocal images of living *pten*-null cells expressing DdPTEN-Halo labeled with TMR and GFP-Nodulin (left) or $PH_{PKB}$-eGFP (right). **c** Confocal images of living *pten*-null cells expressing DdPTEN-Halo (left), DdPTEN$_{G129E}$-Halo (middle) or HsPTEN-Halo (right) and $PH_{PKB}$-eGFP. **d** Quantification of PIP3 levels in the cells in **c** by western blot with anti GFP antibody. The means ± SDs of the ratio of the supernatant fraction to the whole-cell lysate from 3 independent experiments are shown. $P = 0.25$ and 0.06 (DdPTEN versus DdPTEN$_{G129E}$ and HsPTEN) by Welch's *t* test. **e** The pseudopod projection angles from the cells respective to the chemoattractant source (red triangle). Cumulative numbers of membrane protrusion events counted in 15 min in more than 3 cells are shown. **f** Migration velocity. The box, green and black lines, whiskers and number represent the lower and upper quartiles, mean and median, minimum and maximum and the exact maximum value, respectively. The exact *n* values are summarized in Supplementary Table 1. **g**, **j** Confocal images of latrunculin A-treated *pten*-null cells expressing DdPTEN-Halo (left), DdPTEN$_{G129E}$-Halo (middle) or HsPTEN-Halo (right) and $PH_{PKB}$-eGFP in the absence (**g**) or presence (**j**) of 4 mM caffeine. Time, min (**g**) and min:sec (**j**). **h**, **k** Kymographs of the fluorescence intensities measured along the periphery of the cells in **g**, **j**, respectively. The intensities were normalized to those measured in the cytoplasm and shown with the indicated color scale. **i**, **l** Time series of the fluorescence intensities of TMR (black) and eGFP (green) on the periphery of the cells in **g**, **j**, respectively. **m** Autocorrelation functions calculated from **k**. **n** Spatial distribution of DdPTEN-Halo (left) or HsPTEN-Halo (right) and $PH_{PKB}$-eGFP along the cell periphery. The position showing maximum $PH_{PKB}$-eGFP intensity was adjusted to angle = 0. Error bars, SD ($n = 5$ cells). **o** Oscillation period of the traveling waves of $PH_{PKB}$-eGFP in *pten*-null cells expressing DdPTEN-Halo ($n = 94$ cells; black) or HsPTEN-Halo ($n = 67$ cells; gray). Scale bars, 10 μm (**a**–**c**), 5 μm (**g**) and 3 μm (**j**)

caffeine, which inhibits intercellular signaling via cAMP[15]. After the treatment, the PIP3-enriched domain exhibited traveling wave dynamics in about 89% of DdPTEN-expressing cells ($n = 106$ cells), in which the continuous PIP3-enriched domain was displaced perpetually to neighboring positions on the membrane (Fig. 1j, left; Supplementary Movie 4). As shown in a representative kymograph, PIP3 enrichment and DdPTEN enrichment alternated in regular oscillations at every local position along the membrane with the phase of the oscillation gradually changed (Fig. 1k, l, left; Supplementary Fig. 1c). An autocorrelation of the fluorescence intensity time series yielded an oscillation period of 95 ± 38 s (mean ± SD, $n = 94$ cells), and the mean of the fluorescence intensity of the spatial distribution had a half width of 62 ± 11 degrees (mean ± SD, $n = 5$ cells) (Fig. 1m, n, left, o). In DdPTEN$_{G129E}$-expressing cells, no such traveling wave was observed ($n = 46$ cells) (Fig. 1j–m, middle; Supplementary Fig. 1d; Supplementary Movie 5). In contrast, in HsPTEN-expressing cells, a traveling wave was observed not in HsPTEN-Halo-TMR but in $PH_{PKB}$-eGFP in about 64% of the cells ($n = 105$ cells) (Fig. 1j–m, right; Supplementary Fig. 1e; Supplementary Movie 6). The oscillation period of PIP3 was 148 ± 35 s ($n = 67$ cells), and the half width of the distribution was 104 ± 9 degrees ($n = 5$ cells), indicating longer temporal and spatial correlation than in DdPTEN-expressing cells (Fig. 1n, right, o). These findings indicate that the clear spatial separation between PIP3 and DdPTEN is achieved by membrane-localized PTEN through discretization of the PIP3 distribution, which thereby ensures directionally restricted pseudopod projections.

**Bistability of PIP3-enriched and PTEN-enriched states**. We next analyzed the spatial distribution of PTEN by manipulating the PIP3-producing activity of PI3K with a combination of genetic and pharmacological means. The over-expression of myristoylation-tagged PI3K2 (myrPI3K2) in wild-type cells led to a constitutive increase of PIP3 on the whole-cell membrane and the localization of DdPTEN mainly to the cytoplasm (Supplementary Fig. 3a, b)[10]. The enrichment of $PH_{PKB}$-eGFP and DdPTEN-Halo-TMR at a local position on the membrane was quantified by measuring the fluorescence intensities of these two molecules in an approximately 0.03-μm$^2$ ROI normalized to the mean intensities in the cytoplasm, which are designated [PIP3] and [PTEN] hereafter, respectively. The quantification along the periphery of 5 representative cells yielded scatter plots in a PTEN-PIP3 plane exhibiting mainly a single peak at ([PTEN], [PIP3]) = (0.7, 1.5) (Supplementary Fig. 3b). Upon the inhibition of PI3K by adding 40 μM LY294002, PIP3 was cleared due to its

dephosphorylation from the cell membrane where DdPTEN was recruited and accumulated within 3 min (Fig. 2a, top panel; Supplementary Fig. 3a). The distribution of the scatter plots was shifted to a discrete distribution with a single peak at ([PTEN], [PIP3]) = (1.4, 0.9) (Fig. 2b, top panel). [PIP3] = $a$[PTEN], with $a$ equaling ~1.16, formed a boundary between the two distributions (magenta lines in Fig. 2b; Supplementary Fig. 3b, c). We set 40 μM as the highest concentration of LY294002 and serially diluted the concentration to 1 μM. In the representative cell shown in Fig. 2a, multiple steps of dilution reaching 16 μM hardly affected the fluorescence intensities on the membrane. After LY294002 was diluted from 16 μM to 8 μM, the intensities of $PH_{PKB}$-eGFP and DdPTEN-Halo-TMR increased and decreased almost simultaneously in a restricted region on the cell membrane, respectively. Further dilutions to 2 μM gradually expanded the region to cover the whole-cell membrane. All cells showed the conversion from PIP3 enrichment to PTEN enrichment at concentrations ranging from 2 to 16 μM, and the scatter plots exhibited transitions from one distribution to the other without loitering along the boundary (Fig. 2b, top panel). On average, the amplitudes of the two distributions exhibited complementary, ultrasensitive changes in response to PI3K activity (Fig. 2c, Supplementary Fig. 3d, left panels). Every local position on the cell membrane adopted either a PTEN-enriched/PIP3-excluded or PIP3-enriched/PTEN-excluded state, suggesting the characteristics of bistability.

Such bistability in local PIP3 and PTEN levels was not observed in *pten*-null cells expressing DdPTEN$_{G129E}$ or HsPTEN. DdPTEN$_{G129E}$-expressing *pten*-null cells exhibited co-localization of PTEN and PIP3 on the whole-cell membrane almost uniformly and insensitively to PI3K activity (Fig. 2a, Supplementary Fig. 3b, d, middle panels). The scatter plots exhibited constant single-peaked distributions with low positive correlation in the PTEN-PIP3 plane (Fig. 2b, middle panel). In contrast, HsPTEN-expressing *pten*-null cells exhibited responses to not PTEN but PIP3 levels following PI3K manipulation (Fig. 2a, Supplementary Fig. 3b, d, bottom or right panels). PIP3 was accumulated on the whole-cell membrane in the absence of LY294002, but was cleared by dephosphorylation upon treatment with 40 μM LY294002. During serial dilution, PIP3 gradually increased on the whole membrane. The distribution with a single peak at ([PTEN], [PIP3]) = (0.4, 0.8) gradually slid to a distribution with a single peak at ([PTEN], [PIP3]) = (0.4, 2.0) across the boundary of [PIP3] = 1.26 without showing bimodality (Supplementary Fig. 3c, right panel). Therefore, the discretization of local PIP3 levels requires the tightly coupled regulation of reciprocal PTEN membrane localization.

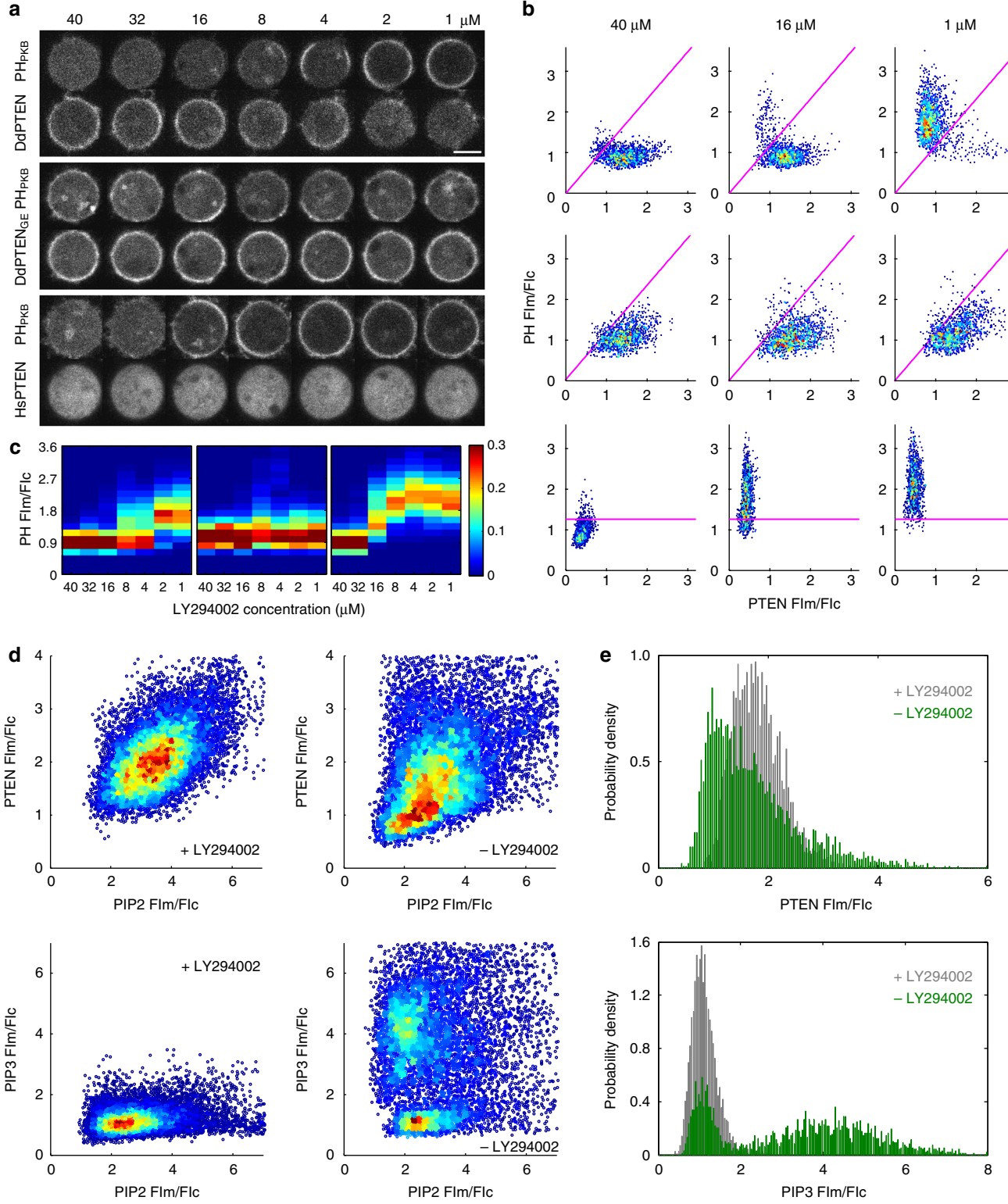

To assess the involvement of PIP2 in the bistable system, the level of which might change during the PIP3 manipulation, scatter plots in PIP2-PTEN and PIP2-PIP3 planes were obtained by their simultaneous measurements in myrPI3K2-expressing cells. In the presence of LY294002, the scatter plot in the PIP2-PTEN plane exhibited a distribution with a single peak at ([PIP2], [PTEN]) = (3.5, 1.7) with positive correlation (Fig. 2d, upper left). In the absence of LY294002, a new distribution with a single peak at ([PIP2], [PTEN]) = (2.8, 1.0) arose, showing that PI3K activity caused a loss of PTEN membrane localization to a larger extent than expected from the positive correlation (Fig. 2d, upper right). Thus, the PIP2 level was no longer a sole determinant for the PTEN level, and an arbitrary [PIP2] value ranging approximately from 2 to 2.9 returned [PTEN] to a value that obeys a bimodal distribution, indicating bistability in the PTEN level (Fig. 2e, upper panel). Within the same range of [PIP2], bistability in the PIP3 level emerged and depended on PI3K activity (Fig. 2d, e, lower panels). These findings suggest that the bistability of the

**Fig. 2** Bistability of PIP3 enrichment tightly coupled to reciprocal PTEN membrane localization. **a** Confocal images of the PTEN-Halo variants labeled with TMR and PH$_{PKB}$-eGFP in living cells during serial dilution of LY294002. Scale bar, 5 μm. **b** Quantification of **a**. Fluorescence intensities of PTEN-Halo-TMR and PH$_{PKB}$-eGFP in a small ROI on the cell membrane were measured along the cell periphery in 5 cells at the indicated LY294002 concentrations. The intensities are normalized to mean fluorescence intensity in the cytoplasm. Higher frequencies are shown in hotter colors. The oblique and horizontal lines show the border between two distributions at the highest and lowest inhibitor concentrations in the DdPTEN and HsPTEN-expressing cells, respectively. **c** Heat maps showing the frequency of PH$_{PKB}$-eGFP intensity at the indicated LY294002 concentrations obtained from **b**. **d** Scatter plots showing fluorescence intensities of GFP-Nodulin and DdPTEN-Halo-TMR (upper) and GFP-Nodulin and PH$_{PKB}$-RFP (lower) in a small ROI along the cell periphery in the presence (left; $n = 25$ and 27 cells for DdPTEN and PH$_{PKB}$-RFP) or absence (right; $n = 28$ and 23 cells) of 40 μM LY294002. Higher frequencies are shown in hotter colors. **e** Histograms of fluorescence intensities of DdPTEN-Halo-TMR (upper) and PH$_{PKB}$-RFP (lower) under conditions where the GFP-Nodulin fluorescence intensity is within the range of 2.0 to 2.9 in the presence (gray) or absence (green) of 40 μM LY294002

PTEN-enriched/PIP3-excluded and PIP3-enriched/PTEN-excluded states arises even under constant PIP2 conditions.

**Mutual inhibition between PIP3 and PTEN.** To explore the mechanism underlying the reciprocal membrane localization between PIP3 and PTEN, we tested the hypothesis that DdPTEN membrane localization is negatively regulated by PIP3 by using other methods to manipulate PIP3 levels. The expression of constitutively active forms of small G protein (RasG$_{Q61L}$, RasC$_{Q62L}$ or Rap1$_{G12V}$) led to an increase in PIP3 levels, such that in some cases the whole membrane was covered with PH$_{PKB}$-eGFP with slight changes in PIP2 levels[21,29,30] (Fig. 3a; Supplementary Fig. 4a-c). In all situations, irrespective of the manipulation method, DdPTEN was excluded from the PIP3-enriched membrane. Treatment with 40 μM LY294002 suppressed the PIP3 enrichment and recovered the membrane localization of DdPTEN (Fig. 3a, middle panel). The exclusion of DdPTEN from the cell membrane did not occur in pi3k1-5-null cells even in the presence of constitutively active small G proteins (Fig. 3a, bottom panel). Therefore, PIP3 triggers the exclusion of DdPTEN from the cell membrane. DdPTEN$_{G129E}$ was also excluded from the PIP3-enriched membrane after the same manipulations as above in wild-type cells expressing endogenous PTEN, suggesting that negative regulation of the phosphatase-dead DdPTEN$_{G129E}$ can be rescued by intact DdPTEN (Fig. 3b; Supplementary Fig. 4d). Both DdPTEN and DdPTEN$_{G129E}$ showed similar kinetics of exclusion with the kinetics of PIP3 enrichment after washing LY294002, reaching a plateau within 3 min (Fig. 3c, d). PIP3-induced fall-off from the membrane was also detected biochemically (Supplementary Fig. 4e). Therefore, the membrane localization of DdPTEN is negatively regulated by PIP3, which, together with the PIP3 dephosphorylation catalyzed by PTEN, constitutes a mutually inhibitory relationship between PTEN and PIP3.

Note that DdPTEN$_{G129E}$ was often localized to the PIP3-enriched membrane in myrPI3K2-expressing cells without LY294002 treatment (Fig. 3b). The probability of co-localization between PTEN and PIP3 was quantified in a scatter plot of their fluorescence intensities (Supplementary Fig. 4f). The probability given by the integral of the probability density from 1.1 to 4.0 for [PTEN] and from 1.375 to 5.0 for [PIP3] was 0.25 when measured between DdPTEN$_{G129E}$ and PIP3 and 0.08 between DdPTEN and PIP3. Therefore, the negative regulation requires both phosphatase-independent and -dependent pathways.

**Single-molecule analysis of PTEN exclusion by PIP3.** To reveal the molecular reactions underlying the negative regulation, the membrane-association and -dissociation kinetics of PTEN were quantified by single-molecule imaging in myrPI3K2-expressing wild-type cells in the presence and absence of 40 μM LY294002[31–33] (Fig. 4; Supplementary Fig. 5; Supplementary Movie 7–9). Nonspecific binding of TMR-HaloTag ligand to

D. discoideum cells was negligible under our experimental conditions (Supplementary Fig. 5a). Proof for single-molecule imaging of PTEN-Halo-TMR includes multiple features of the visualized fluorescent spots, such as an almost constant appearance, constant fluorescence intensity and single-step photobleaching (Supplementary Fig. 5c, d). The time durations of the membrane binding were measured for each DdPTEN molecule detected in the presence (low PIP3) or absence (high PIP3) of LY294002 (Fig. 4b; Supplementary Fig. 5b). The dissociation curves showed faster dissociation from the high PIP3 membranes (Fig. 4c, dark green) than from the low PIP3 membranes (Fig. 4c, pale green). Each of the dissociation curves was best fitted with three exponential components, and the averaged membrane-binding lifetime of DdPTEN was 4.6 s and 1.7 s on the low and high PIP3 membranes, respectively, suggesting 2.7-fold acceleration of the membrane dissociation by PIP3 (Table 1). The averaged membrane-binding lifetime of DdPTEN$_{G129E}$ was 3.5 s and 2.3 s on the low and high PIP3 membranes, respectively, showing 1.5-fold acceleration. Therefore, the substrate PIP3 lowers the affinity of enzymatically active PTEN for the membrane.

PIP3 also affected the membrane-association frequency. The number of PTEN-Halo-TMR molecules appeared on the membrane per unit time interval per unit area was measured in multiple cells having different fluorescence intensities of cytoplasmic PTEN-Halo-TMR. The scatter plot of the number of molecules against the intensity showed linear regression, with the slope representing the relative frequency of membrane associations between PTEN-Halo-TMR and the membrane (Fig. 4d; Supplementary Fig. 5e). The frequencies obtained between DdPTEN and DdPTEN$_{G129E}$ and high and low PIP3 membranes revealed that PIP3 suppresses the membrane association of DdPTEN irrespective of the phosphatase activity. Taken together, membrane association is suppressed and dissociation is promoted by PIP3 irrespective of the enzyme activity, and membrane dissociation is further promoted by the dephosphorylation of PIP3.

**Suppression of PTEN's stable membrane binding by PIP3.** Three membrane-binding states, which possibly arose due to differences in the molecular entity of the binding site in the membrane, have been found in DdPTEN$_{G129E}$-Halo-TMR previously[25]. Therefore, we applied our analysis to wild-type DdPTEN as well as DdPTEN$_{G129E}$ under the same high and low PIP3 conditions[34]. Three diffusion coefficients were estimated to be 0.01–0.03, 0.05–0.08 and 0.43–0.75 μm²sec⁻¹ for stably, primary and weakly binding states, respectively, and none were significantly affected by either the PTEN variations or PIP3 levels (Table 1). The fraction of molecules adopting the stably, primary and weakly binding states was 41–44%, 50–56% and 3–6% on the high PIP3 membrane and 60–67%, 30–35% and 3–5% on the low PIP3 membrane, respectively. Since the total

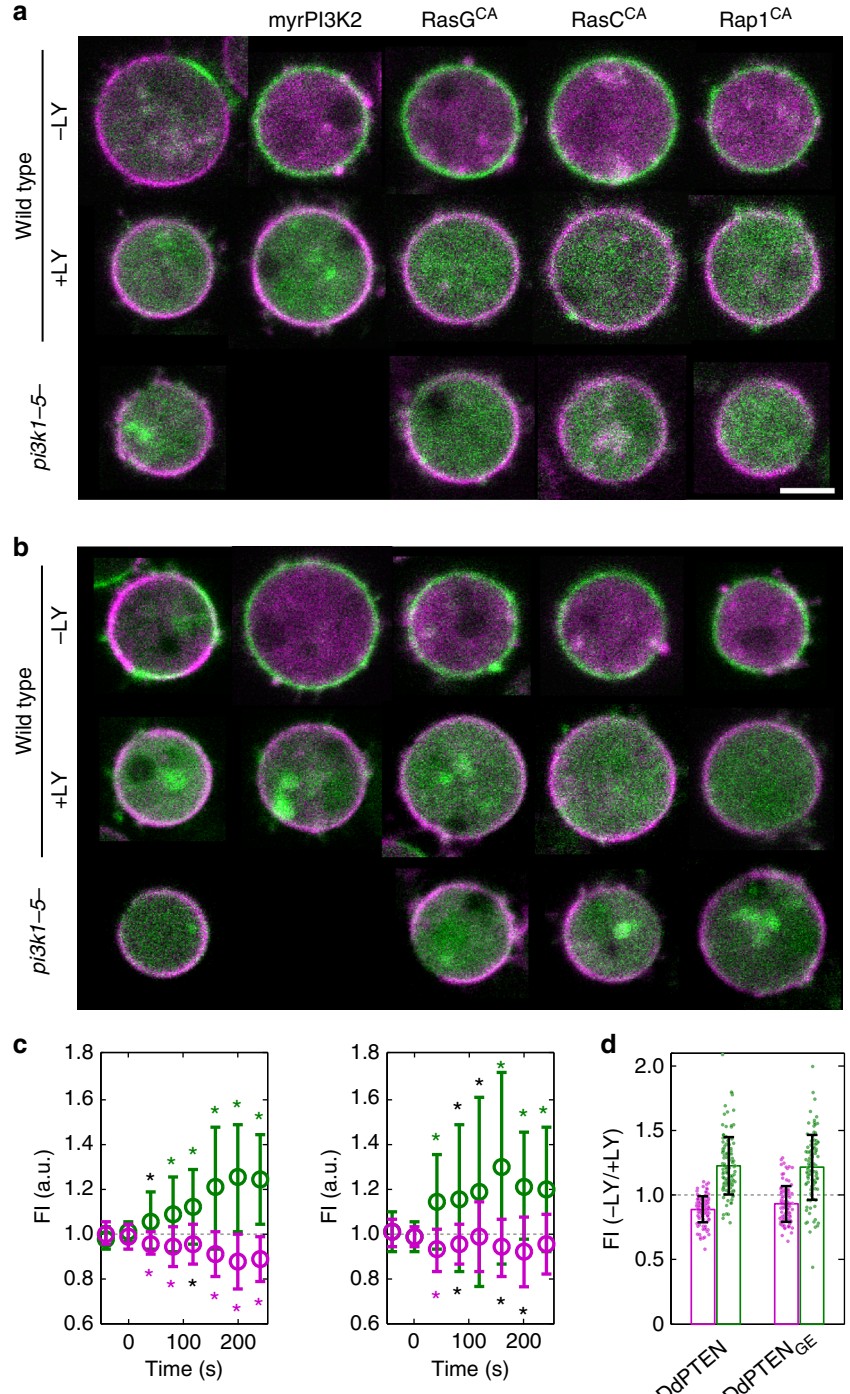

**Fig. 3** Mutual inhibition between PIP3 and PTEN excludes each other from the membrane. **a** Confocal images of DdPTEN-Halo-TMR and PH$_{PKB}$-eGFP in living cells expressing myrPI3K2 or constitutively active forms of small G proteins in the absence (top) or presence (middle) of 40 μM LY294002 or in the background of $pi3k1$-$5$-null (bottom). **b** Confocal images of DdPTEN$_{G129E}$-Halo-TMR and PH$_{PKB}$-eGFP in living cells expressing myrPI3K2 or constitutively active form of small G proteins. **c** Quantification of the fluorescence intensities of DdPTEN-Halo (open magenta circle; left; $n = 17$) or DdPTEN$_{G129E}$-Halo (right; $n = 15$) and PH$_{PKB}$-eGFP (open green circle) on the cell membrane of wild-type cells expressing myrPI3K2 before and after the dilution of LY294002 at $t = 0$. Error bars, SD. Black and colored asterisks indicate significant differences after the dilution ($P < 0.05$, $P < 0.01$). **d** Fluorescence intensities of DdPTEN (magenta) and PH$_{PKB}$-eGFP (green) after the dilution relative to before the dilution. The data obtained by 7 measurements at different time points in the same cells as **c** were shown. Error bars, SD. $P = 0.012$ and $0.742$ (DdPTEN versus DdPTEN$_{G129E}$ for DdPTEN and PH$_{PKB}$). $P$ values were obtained by Welch's $t$ test. Scale bar, 5 μm

number of membrane-bound PTEN molecules was reduced 0.73–0.80 fold by the PIP3 increase (Supplementary Fig. 4e), the number of molecules adopting the stably, primary and weakly binding states on the high PIP3 membrane was equivalent to 32–33%, 37–45% and 2–4% of the total number of membrane-bound PTEN molecules on the low PIP3 membrane (Fig. 5a). The specific reduction of molecules adopting the stably binding state on the high PIP3 membrane suggests inactivation of the

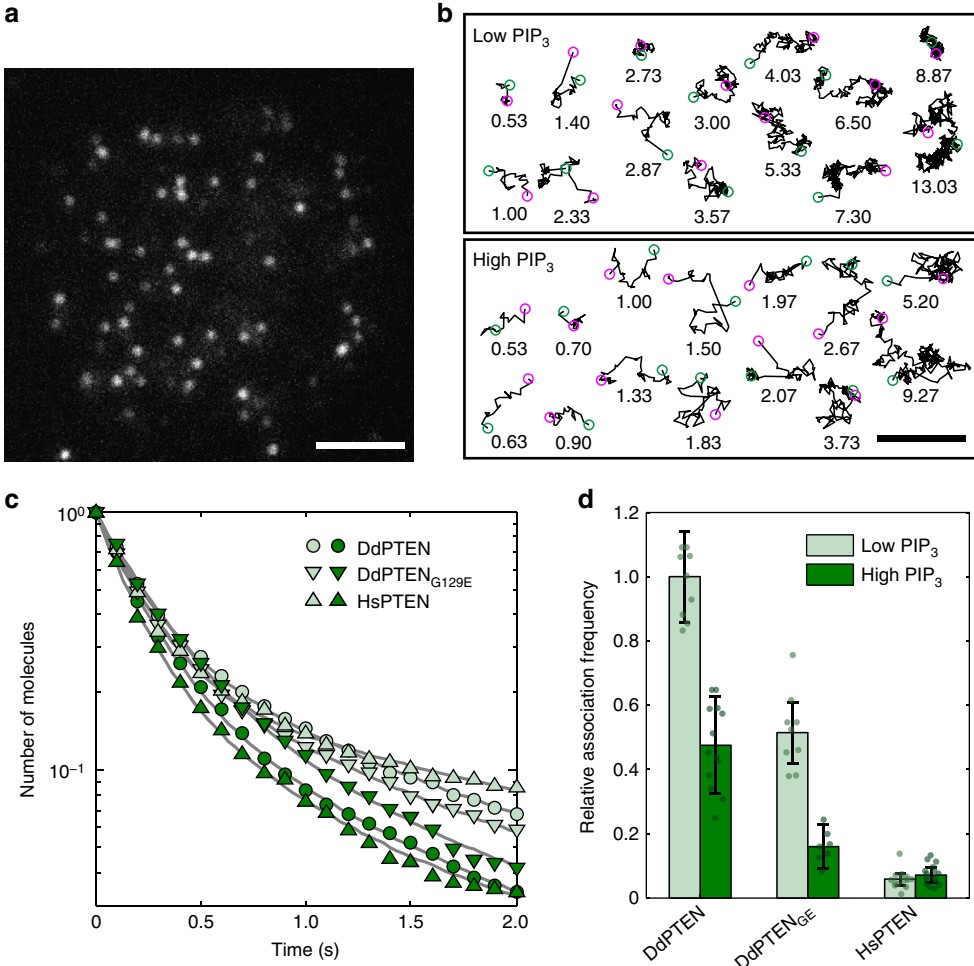

**Fig. 4** PIP3 modulates the membrane-association and -dissociation kinetics of PTEN. **a** A TIRF image of DdPTEN-Halo-TMR in a living cell expressing myrPI3K2 in the presence of 5 μM latrunculin A and 40 μM LY294002. **b** Trajectories of single DdPTEN molecules undergoing lateral diffusion on the cell membrane of myrPI3K2-expressing cells in the presence (upper, low PIP3) or absence (lower, high PIP3) of LY294002. Start and end positions are indicated by the magenta and green circles, respectively. Time, sec. **c** Dissociation curves of DdPTEN (circle), DdPTEN$_{G129E}$ (inverted triangle) and HsPTEN (triangle) from the low (pale green) and high (dark green) PIP3 membranes. Lines represent fitting to equation (1). The data are represented as means, and statistics including the exact *n* values are summarized in Table 1. **d** Relative association frequencies of DdPTEN (*n* = 10 and 14 cells), DdPTEN$_{G129E}$ (*n* = 12 and 7 cells) and HsPTEN (*n* = 15 and 15 cells) to the low (pale green) and high (dark green) PIP3 membranes, respectively. The actual data are calculated accordingly to the regression equation. Error bars, s.e. *P* = 0.085, 0.159, and 0.400 (low PIP3 vs. high PIP3 for DdPTEN, DdPTEN$_{G129E}$, and HsPTEN, respectively). *P* = 0.071 and 0.000 (DdPTEN versus DdPTEN$_{G129E}$ and DdPTEN versus HsPTEN, respectively, under low PIP3 conditions). *P* values were obtained by regression slope test. Scale bars, 1 μm

corresponding binding site by PIP3, although the molecular entity remains to be identified.

The mechanism for how the suppression of the stably binding state by PIP3 promotes membrane dissociation was investigated by analyzing the kinetics of the membrane-binding state transitions, as described previously[25,35]. We classified the states of membrane-bound molecules every 33 msec after the onset of membrane association. For simplicity, the three states were classified into two states, a stably binding state and a transiently binding state, which was defined as a mixture of the primary and weakly binding states. Some molecules dissociated from the membrane and some changed their membrane-binding state, resulting in the states reaching steady state within 1 s (Fig. 5b). Rate constants of the membrane dissociation and state transition were estimated from the kinetics (Fig. 5c). The estimated dissociation rate constants showed that PTEN hardly dissociated from the membrane when it adopted the stably binding state. Thus, the PIP3-induced reduction of the

stably binding state accelerated the membrane dissociation irrespective of the lipid phosphatase activity. While the state transitions were not significantly affected by mutations on low PIP3 membranes, DdPTEN exhibited an increase in the rate constant of the stably-to-transiently binding state transition due to the PIP3 enrichment, whereas DdPTEN$_{G129E}$ did not. Since the G129E mutation reduces the cavity of the catalytic site and prevents substrate binding[36], PIP3 bound to the catalytic site would seem to release PTEN from the stably binding state and thus promote membrane dissociation. Taken together, freely diffusing PIP3 inactivates the binding site required for stable binding to the membrane, thus suppressing membrane association and promoting membrane dissociation. PIP3 bound to the catalytic site of PTEN also reduces the affinity between the binding site and PTEN, which further promotes membrane dissociation. These findings provide a mechanism for the negative regulation of PTEN membrane localization that depends on local PIP3 levels (Figs. 5e, 6a).

**Table 1 Single-molecule measurements of membrane-dissociation kinetics and diffusion coefficients**

|  | DdWT/ + LY | DdWT/ − LY | DdGE/ + LY | DdGE/ − LY | HsWT/ + LY | HsWT/ − LY |
|---|---|---|---|---|---|---|
| *Dissociation* |  |  |  |  |  |  |
| $k_1$ [sec$^{-1}$] | 0.11 (n.d.) | 0.22 (0.10) | 0.13 (0.01) | 0.18 (0.05) | 0.13 (0.00) | 0.32 (0.20) |
| $k_2$ [sec$^{-1}$] | 1.04 (0.92) | 1.37 (1.25) | 1.05 (0.93) | 1.36 (1.23) | 0.63 (0.51) | 1.85 (1.73) |
| $k_3$ [sec$^{-1}$] | 4.82 (4.69) | 5.01 (4.88) | 5.06 (4.93) | 4.14 (4.01) | 4.79 (4.66) | 6.30 (6.17) |
| $q^k_1$ | 0.04 | 0.03 | 0.04 | 0.04 | 0.05 | 0.05 |
| $q^k_2$ | 0.28 | 0.22 | 0.25 | 0.27 | 0.17 | 0.25 |
| $q^k_3$ | 0.68 | 0.75 | 0.71 | 0.69 | 0.79 | 0.70 |
| Residual | 0.0060 | 0.0078 | 0.0036 | 0.0070 | 0.0066 | 0.0054 |
| Lifetime [sec] | 4.62 | 1.72 | 3.49 | 2.29 | 4.25 | 1.43 |
| *Diffusion* |  |  |  |  |  |  |
| $D_1$ [μm$^2$ sec$^{-1}$] | 0.01 | 0.01 | 0.01 | 0.03 | 0.02 | 0.02 |
| $D_2$ [μm$^2$ sec$^{-1}$] | 0.05 | 0.05 | 0.06 | 0.08 | 0.07 | 0.14 |
| $D_3$ [μm$^2$ sec$^{-1}$] | 0.52 | 0.43 | 0.75 | 0.66 | 0.53 | 0.77 |
| $q^D_1$ | 0.67 | 0.41 | 0.60 | 0.44 | 0.66 | 0.40 |
| $q^D_2$ | 0.30 | 0.56 | 0.35 | 0.50 | 0.26 | 0.41 |
| $q^D_3$ | 0.03 | 0.03 | 0.05 | 0.06 | 0.08 | 0.19 |
| *Data number* |  |  |  |  |  |  |
| Molecules | 1322 | 1280 | 1318 | 1490 | 781 | 972 |
| Cells | 20 | 13 | 19 | 19 | 24 | 16 |

The parameters $k_{1/2/3}$ and $q^k_{1/2/3}$ were estimated by fitting the dissociation curve to equation (1) with the least-squares method. Squared 2-norms of the residuals are indicated. The parameters $D_{1/2/3}$ and $q^D_{1/2/3}$ were estimated by fitting the displacement distribution to equation (3) with the maximum likelihood estimation method. Numbers in parenthesis represents the rate constants after subtraction of the photo-bleaching rate constant

**Regulation of the area and location of PIP3 enrichment**. To reveal the molecular reactions essential for steep PIP3 enrichment, the single-molecule behaviors of HsPTEN were analyzed under the same high and low PIP3 conditions (Figs. 4, 5 and Table 1). Once bound to the membrane, the diffusion coefficients and the kinetics of both the membrane dissociation and state transitions of HsPTEN were very similar to those of DdPTEN, including their sensitivity to PIP3. However, the membrane association was significantly slow compared to DdPTEN irrespective of PIP3 levels. This difference is because the membrane-binding surface of cytoplasmic HsPTEN is covered with the phosphorylated C-terminal tail[24,37,38]. As a consequence, an enhancement of association to the low-PIP3 membrane was not observed even if the binding site for the stably binding state was released from the inactivation. Therefore, a sufficient amount of PTEN is necessary at the low PIP3 membrane to confine the PIP3-enriched domain (Fig. 6b).

Finally, we examined the effects of a chemoattractant, cAMP, on the membrane-binding states in *pi3k1-5*-null cells, where no PIP3 was detected even after cAMP stimulation[28]. As shown in Fig. 5d, cAMP stimulations transiently caused a reduction in the amount of membrane-bound DdPTEN and suppression of the stably binding state. Similarly, the amount of membrane-bound DdPTEN$_{G129E}$ was reduced by cAMP stimulation in *pten*-null cells without detectable changes in PIP3 levels (Supplementary Fig. 2e). Thus, the stably binding state of PTEN is suppressed by both PIP3 and some other signal(s) derived from the chemoattractant, leading to PTEN dissociation from the membrane. Therefore, the PIP3-enriched domain arises due to the suppression of the stably binding state in response to an externally applied cAMP gradient as well as intrinsically fluctuating PIP3 for cell migration (Figs. 5e, 6c).

## Discussion

PTEN and PIP3 are two molecules that antagonistically polarize on the cell membrane during cell motility. We show that the mutual inhibition between PTEN and PIP3 generates bistability in the PIP3-enriched/PTEN-excluded and PTEN-enriched/PIP3-excluded states. Bistability allows a biological system to adopt one of two metastable states with different molecular activities, concentrations or densities. Interchanges between the states take place quickly in an ultrasensitive manner, often resulting in a switch-like behavior in the downstream signaling[26,39]. We found that PI3K activity determines which state is chosen (Fig. 2b). At moderate PI3K activity, both the PIP3-enriched and PTEN-enriched states appeared separately in space, resulting in two discrete domains. Abrogation of the bistability by replacing DdPTEN with HsPTEN or DdPTEN$_{G129E}$ allowed intermediate PIP3 enrichment, which slowed the dynamics of the domain border or caused a uniform distribution of PIP3 across the whole membrane. Thus, bistability arising from the mutual inhibition provides a mechanistic basis for the discreteness of the PIP3-enriched domain and clear spatial separation between the anterior and posterior signals.

The loss of bistability compromised the confinement of PIP3, as seen by the projection of lateral and posterior pseudopods that in turn disrupted cell migration (Fig. 1c–f; Supplementary Movies 1-3; Supplementary Fig. 2). Thus, bistability enforces efficient directed cell migration by discretizing the spatial anterior signal. Our results strengthen the idea that PIP3 serves as a signal for pseudopod formation[1,2]. PIP3 signaling contributes to chemotaxis as one of four redundant and parallel signaling pathways downstream of the chemoattractant receptor, each of which with the potential to induce asymmetric motility[40,41]. Indeed, cells without PI3K activity can exhibit chemotaxis toward cAMP rather effectively[28,42]. However, cells without PTEN activity hardly migrate toward the chemoattractant gradient, which questions the redundancy of the signaling pathways. The PIP3 3-phosphatase activity in chemotactically-responding *D. discoideum* is mainly provided by *ptenA*, the target of this study. *ptenA* disruption leads to a loss of PIP3 degradation when it is enhanced after chemoattractant stimulation[5,28], suggesting contributions by other PIP3 3-phosphatase genes are insufficient for chemotactic signaling[43]. Defects of *ptenA*-null cells in chemotaxis can be overcome by suppressing PIP3 signaling with PI3K inhibitors[44]. That is, the PIP3 3-phosphatase activity of PTEN is necessary in the presence of PI3K activity for effective chemotaxis. Without the antagonization against the anterior signal by the posterior enzyme, the signal disperses across the membrane and disturbs

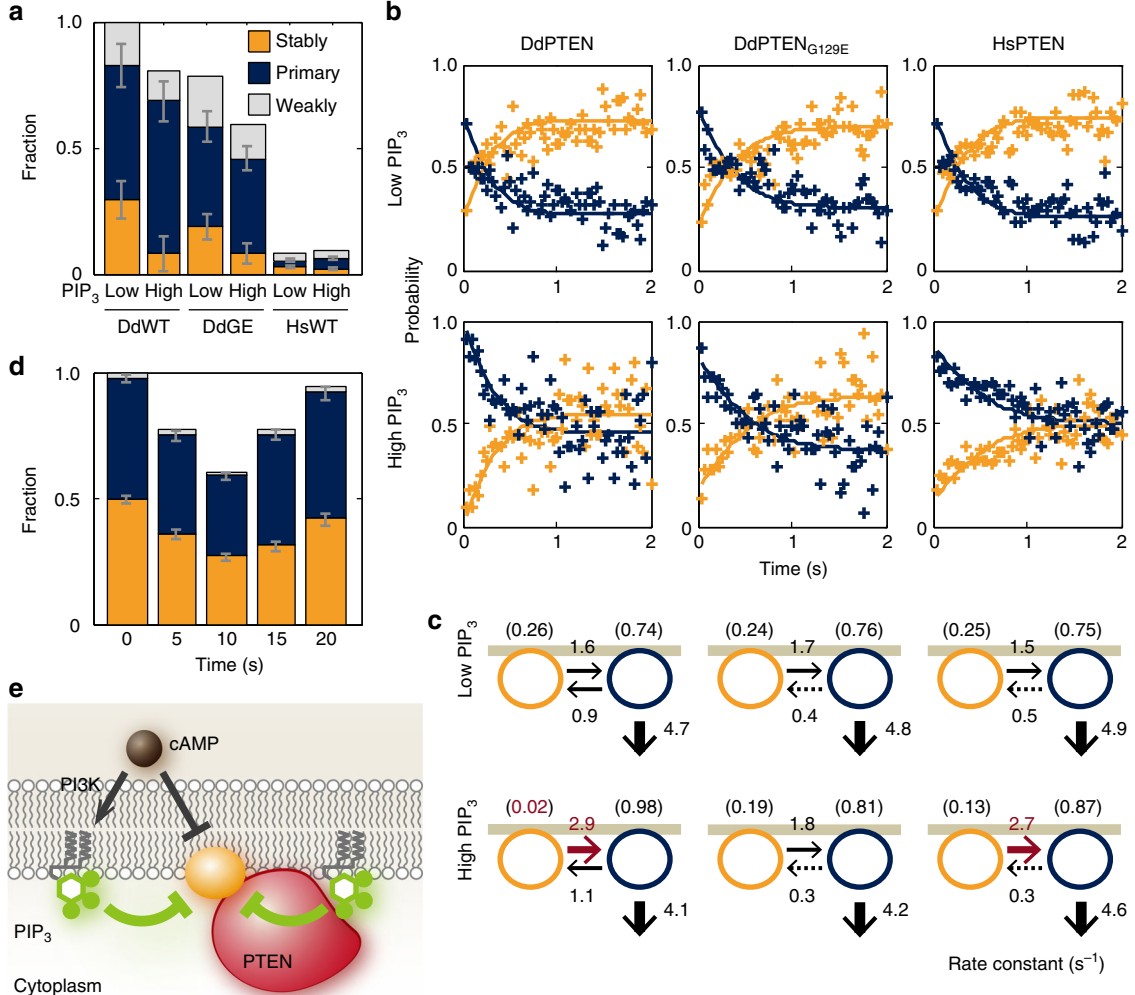

**Fig. 5** Negative regulation of PTEN membrane localization through binding sites sensitive to PIP3. **a** Three diffusion states of DdPTEN, DdPTEN_G129E and HsPTEN molecules were detected on the low and high PIP3 membranes. The values were normalized to the total number of DdPTEN molecules on the low PIP3 membranes. Error bars: 95% confidence intervals. $P = 0.000$, 0.002, and 0.000 (low PIP3 vs. high PIP3 for DdPTEN, DdPTEN_G129E, and HsPTEN) and 0.000 and 0.000 (DdPTEN vs. DdPTEN_G129E and DdPTEN vs. HsPTEN under the low PIP3 conditions). **b** Stably (orange) and transiently (blue) binding states of membrane-bound molecules displayed in a time series after the onset of membrane binding. The lines represent the fitting to the solutions of the simultaneous differential equations in equation (6). **c** State transitions and membrane-dissociation kinetics. Numbers without parenthesis represent rate constants (sec⁻¹). Numbers in parenthesis represent probabilities of adopting the respective state at the moment of membrane association. **d** Three binding states of DdPTEN molecules undergoing a transient response to uniform cAMP stimulation in *pi3k1-5*-null cells ($n = 3342$ molecules, 16 cells). The values were normalized to the total number of DdPTEN molecules before the stimulation. Error bars: 95% confidence intervals. $P = 0.004$, 0.000, 0.000 and 0.000 (before stimulation vs. 0–5 s, 5–10 s, 10–15 s and 15–20 s after stimulation). $P$ values were obtained by likelihood-ratio test. **e** A model for negative regulation of PTEN membrane localization. It is assumed that a specific binding site (orange circle) acts as the stabilizer that associates with PTEN in the stably bound state and that this binding site is inhibited by PIP3 via PTEN-dependent and –independent mechanisms as well as by cAMP signaling

the anterior–posterior polarity that might be established via the other pathways. Thus, mutual inhibition might be a mechanism between anterior and posterior signals to avoid unconfined enrichment. Bistability arises due to two positive-feedback loops that amplify one of two mutually inhibitory components to produce ultrasensitive interchanges between the metastable states;[45] PIP3 and PTEN are presumably amplified by Ras-PI3K-PIP3 and PTEN-PIP2 positive-feedback loops, respectively[22,23]. Small G proteins and other phosphoinositides may exhibit bistability similarly by constituting mutually inhibitory relationships between two positive-feedback loops for the spatial separation between anterior and posterior signals[46].

This study demonstrates the negative regulation of PTEN membrane localization by PIP3 at the resolution of the molecular ensemble and the individual single molecule. At least two

mechanisms are revealed to be involved in the negative regulation: one independent and the other dependent on the lipid phosphatase activity of PTEN. The phosphatase activity-independent mechanism involves a PIP3-induced reduction of the binding site for PTEN to adopt the stably binding state. The reduction causes not only a reduced frequency of membrane associations but also an increased rate of membrane dissociations, since the binding site interacts with both cytoplasmic and membrane-bound PTEN. The affinity between the binding site and PTEN is likely to be negatively regulated by PIP3 and cAMP signaling activity (Fig. 5a, d, e). The phosphatase activity-dependent mechanism involves the substrate-induced release of PTEN from the stably binding state, and thereby promotes the membrane dissociation. PIP3 bound to the catalytic site lowers the affinity between the binding site and PTEN, an effect that

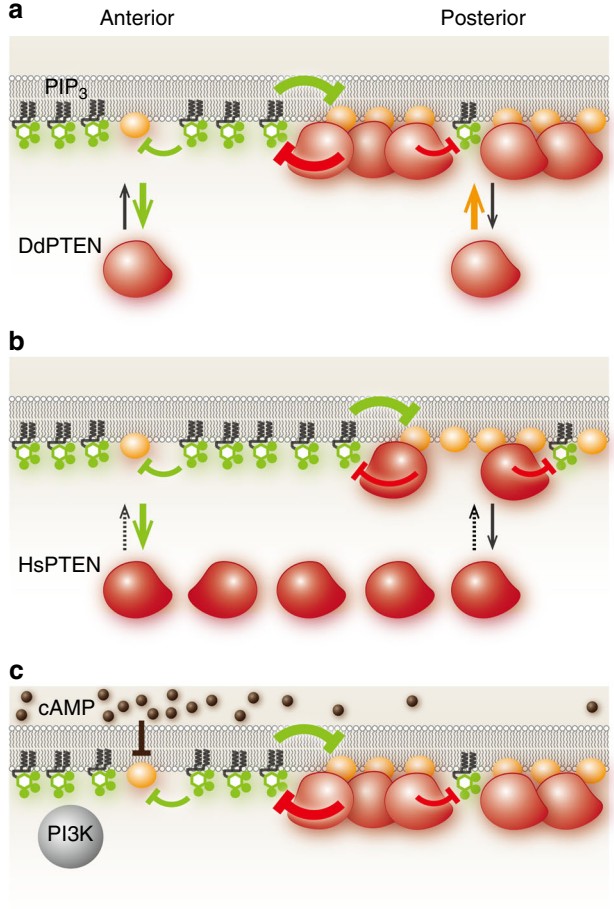

**Fig. 6** A model explaining the molecular mechanism for confining and localizing PIP3 enrichment. **a** Clear spatial separation between DdPTEN and PIP3 established via mutual inhibition in the presence of negative regulation and phosphatase activity. **b** Enlargement of the PIP3-enriched domain by the replacement of DdPTEN with HsPTEN, which lacks sufficient membrane localization. **c** Chemotactic signaling excludes DdPTEN from the higher concentration side by suppressing the binding site for the stably binding state

occurs rapidly on the PIP3-enriched membrane due to the reduced waiting time before PTEN encountering the substrate, PIP3. These two mechanisms are necessary for the mutual inhibition, however, the molecular entity of the binding site has yet to be identified. Determining the number of dephosphorylation reactions catalyzed by a single PTEN molecule per single membrane-binding event in the absence and presence of this unknown molecule will distinguish which of the stably binding or transiently binding states exert stronger enzymatic activity and thus clarify the molecular mechanism for the negative regulation.

Defects in the PTEN variants in the spatial separation of PTEN and PIP3 can be attributed to defects in the negative regulation as follows (Figs. 1, 4, 5). DdPTEN$_{G129E}$ does not exhibit PIP3 dephosphorylation or subsequent substrate-induced release from the stably binding state (Fig. 5c). Thus, the membrane dissociation of DdPTEN$_{G129E}$ is induced less effectively by PIP3 than that of DdPTEN or HsPTEN (Fig. 4c), resulting in the co-localization of PIP3 and PTEN on the membrane (Fig. 1). Since cytoplasmic HsPTEN constitutively adopts the closed conformation, in which the membrane interacting surface is covered with the phosphorylated C-terminal tail, the membrane-association frequency

is low irrespective of the PIP3 level[24,37,38] (Fig. 4d). Thus, a sufficient amount of PTEN to dephosphorylate PIP3 fails to accumulate on the membrane outside the PIP3-enriched domain, resulting in a slowly dynamic domain border (Fig. 1).

The PIP3-enriched domain exhibits traveling waves, which is characteristic of an excitable system that autonomously outputs a stereotypical spike-like response (Fig. 1k)[15,19,46]. Each single event of the excitation is followed by a refractory period, which suppresses the next excitation temporarily[47]. Thus, under a constitutively super-threshold condition of an excitable system, the excitation is expected to be repeated, which could be visualized as a spatiotemporal oscillation such as traveling wave. On the other hand, in the presence of chemoattractant gradients, excitable networks cannot explain continuous PIP3-enriched domains facing the source[7,18,21,48–51]. To generate a continuous PIP3-enriched domain, the chemoattractant receptor signal most likely regulates PI3K and PTEN without any refractory period. To explain the PIP3-enriched domain generation and localization, we propose a model based on a combination of excitable and bistable systems as follows.

Our model assumes that Ras is responsible for the emergence of the excitable dynamics based on evidence that an active form of Ras exhibits traveling waves in the absence of PIP3 traveling waves[52]. The active Ras provides a binding site for PI3K on the cell membrane[4]. Our data suggest that the PI3K activity can determine the state on the cell membrane. That is, the ultrasensitive interchanges between PTEN-enriched/PIP3-excluded and PIP3-enriched/PTEN-excluded states can be induced according to the excitable dynamics of Ras. While the excitation occurs stochastically without any extracellular spatial cues, an asymmetric signal derived from the chemoattractant gradient at the higher concentration side can hijack the bistable system to induce state interchanges by suppressing the binding site of PTEN as well as by activating PI3K[53]. Thus, the PIP3-enriched domain emerges to orient itself toward the chemoattractant source, providing directed motility as long as a gradient exists.

The main role of the chemoattractant gradient is to choose the location where the anterior signaling state is adopted. From a microscopic view, cAMP selectively inactivates the binding site for PTEN to take the stably binding state among three possible membrane-binding states. From a macroscopic view, the chemoattractant gradient restricts the directionality of migration that would otherwise be effectively random without the gradient; i.e. biased Brownian motion[54,55]. Therefore, there is a mechanistic conservation across hierarchical levels of the biological system from the single molecule to the single cell, illustrating that selection among multiple intrinsic states in response to extrinsically applied signals underlies cellular behavior. A bistable switch working at the confluence of the extrinsic and intrinsic signals may be one of the fundamental characteristics of the signaling system for a cell's decision making.

## Methods

**Cell culture.** *Dictyostelium discoideum* wild-type strain Ax2 (dictyBase strain ID: DBS0235518), a kind gift from M. Maeda was used as the parental cell line throughout this study. *pi3k1-5*-null strain (dictyBase strain ID: DBS0252652) was a kind gift from R. Kay (MRC, Cambridge, UK). All cell lines were grown in HL5 medium (15.4 g glucose, 7.15 g yeast extract, 14.3 g proteose peptone No.2, 0.485 g KH$_2$PO$_4$, 1.28 g Na$_2$HPO$_4$12H$_2$O, 0.2 mg folic acid, 0.06 mg cyanocobalamin per L) supplemented with penicillin and streptomycin at 21 °C.

**Construction of cell lines.** A *pten*-null cell line was generated by homologous recombination. The construct for gene knockout was made and consisted of 5′- and 3′-fragments of the *pten* gene (GenBank: AF483827.1) intervened by an expression cassette for the blasticidine resistance gene. The 5′- and 3′- fragments

were amplified from *D. discoideum* genomic DNA by PCR using primers, DdPTEN-KO1 and DdPTEN-KO2, and DdPTEN-KO3 and DdPTEN-KO4 listed in Supplementary Table 2, respectively. The whole construct was amplified by fusion PCR using the amplified fragments and the cassette, which was introduced into *D. discoideum* wild-type Ax2 strain by electroporation. For cell lines expressing constitutively membrane-targeted PI3K, a gene coding for myrPI3K2, in which PI3K2 was N-terminally conjugated with 16 amino acids from chicken c-Src, was amplified by PCR using primers, PI3K2-1, PI3K2-2 and PI3K2-3, and cloned into the AatII and SpeI sites in an extrachromosomal plasmid, pHK12bla[10]. The 16 amino acid sequence was MGSSKSKPKDPSQRRR, which constitutes a myristoylation signal and basic amino acids sufficient for stable membrane association. The construct was introduced into the cells by electroporation. These transformants were selected under 10 µg mL$^{-1}$ Blasticidine S. For the simultaneous expression of PTEN-Halo and PH$_{PKB}$-eGFP, a gene coding for DdPTEN-, DdPTEN$_{G129E^-}$ or HsPTEN-Halo was amplified by PCR using primers, DdPTEN-OE1 and DdPTEN-OE2, and HsPTEN-OE1 and HsPTEN-OE2, and cloned into the BglII and SpeI sites of pDM344[56] (GenBank: FJ402941). In the gene coding for HaloTag protein, the NgoMIV site was eliminated by synonymous substitution that was introduced by site-directed mutagenesis (Stratagene). The NgoMIV-digested fragment of pDM344 consisting of actin15 promoter, PTEN-Halo gene and actin8 terminator was cloned into a NgoMIV site of an extrachromosomal plasmid, pDM181, in which a gene coding for PH$_{PKB}$-eGFP was cloned into the BglII and XbaI sites. The plasmid was introduced into the cells by electroporation, and the transformants were selected under 20 µg mL$^{-1}$ Geneticine. For the doxycycline-induced expression of constitutively active forms of small G proteins, a gene coding for RasG$_{Q61L}$, RasC$_{Q62L}$ or Rap1$_{G12V}$ was cloned into the BglII and SpeI sites of an extrachromosomal plasmid, pDM359 (GenBank: EU908846). The plasmid was introduced into the cells by electroporation, and the transformants were selected under 40 µg mL$^{-1}$ Hygromycin B.

**Cell preparation for microscopy.** Cultured cells were starved before microscopic observation as follows. The cells were washed twice with development buffer (DB; 5 mM Na/KPO$_4$, 2 mM MgSO$_4$, 0.2 mM CaCl$_2$, pH 6.5) by centrifugation at 500 × *g* for 2 min. In the case of myrPI3K2-expressing cell lines, 5 mL of the cell suspension at 1 × 10$^6$ cells mL$^{-1}$ was transferred to a 20 mL flask and shaken at 150 rpm for 4 h at 21 °C. In the case of the other cell lines, 1 mL of the cell suspension at 3 × 10$^6$ cells per mL was transferred to a 35-mm culture dish and kept still for 4 h at 21 °C. During the last 30 min, HaloTag TMR ligand (Promega) was added to the cell suspension at the final concentration of 2 µM and 20 nM for subcellular localization imaging and single-molecule imaging, respectively. After incubation, the cells were washed twice with DB by centrifugation and suspended in DB at around 5 × 10$^6$ cells per mL.

**Cell migration imaging.** A concentration of 10 µL suspension of the starved cells was placed on a coverslip of a 35-mm glass bottom dish (27-mm glass in diameter, Iwaki). The cells were allowed to adhere to the surface by 10-min incubation, filled with 1 mL DB and allowed to settle by another incubation for at least 20 min. The cells were observed under an inverted microscope (IX71, Olympus) equipped with a time-lapse camera (DS-2MBW, Nikon). In order to stimulate the cells with a spatial gradient of cAMP, DB containing 1 mM cAMP filled in a glass micropipette (Femtotips II, Eppendorf) was released by applying 50 hPa using FemtoJet (Eppendorf). The images were acquired at a time interval of 1 s or 5 s for 30 min.

**Subcellular localization imaging.** To observe the spontaneous dynamics of PTEN and PIP3, 200–300 µL suspension of starved cells in DB containing 5 µM latrunculin A (Sigma) and 4 mM caffeine was placed on a coverslip of a 35-mm glass bottom dish (12-mm glass in diameter, Iwaki). The cells were allowed to settle by 30-min incubation and then observed under laser confocal microscopy (A1, Nikon or FV-1000, OLYMPUS). The images were acquired at a time interval of 5 s for 30 min.

To observe the response to the manipulation of PI3K activity, 15 µL suspension of the starved cells in DB containing 10 µM latrunculin A was placed on a coverslip of a 35-mm glass bottom dish (12-mm glass in diameter, Iwaki). The cells were allowed to settle by 30-min incubation and then observed under laser confocal microscopy. Time-lapse imaging was started, and at the same time 5 µL DB containing 10 µM latrunculin A and 160 µM LY294002 (Cayman Chemical) was added to the cell suspension to treat the cells with LY294002 at a final concentration of 40 µM. Every 3 min, 5, 25, 50, 100, 200, or 400 µL of DB containing 10 µM latrunculin A was added to the cell suspension to serially dilute LY294002 to 32, 16, 8, 4, 2, and 1 µM.

To observe PTEN and PIP3 in the cell lines expressing constitutively active forms of small G proteins, 300 µL suspension of the starved cells in DB containing 10 µM latrunculin A was placed on a coverslip of a 35-mm glass bottom dish (12-mm glass in diameter, Iwaki). The cells were allowed to settle by 30-min incubation. Then, 100 µL DB containing 10 µM latrunculin A and 40 µg mL$^{-1}$ doxycycline (doxycycline hydrochloride, MP Biomedicals) was added, and the cells were incubated for another 45 min and finally observed under laser confocal microscopy.

To observe PTEN and PIP3 in response to a chemoattractant gradient, 300 µL suspension of the starved cells in DB containing 5 µM latrunculin A was placed on a coverslip of a 35-mm glass bottom dish (12-mm glass in diameter, Iwaki). The cells were allowed to settle by 10-min incubation. Then, 2000 µL DB containing 5 µM latrunculin A was added, and the cells were incubated for another 20 min and finally observed under laser confocal microscopy. DB containing 10 µM cAMP filled in a glass micropipette (Femtotips, Eppendorf) was released by applying 50 hPa using FemtoJet. The images were acquired every 5 s, and 12 images after 1–2 min of gradient application was averaged and used for the analysis.

**Single-molecule imaging.** A 10 µL suspension of the starved cells was placed on a coverslip that was cleaned by sonication in 0.1 N KOH for 30 min and washed alternately with 99.5% EtOH (special grade) and distilled water twice prior to use. The cells were allowed to adhere to the surface by incubating the coverslip in a moist chamber for 10 min. The cells were overlaid with an agarose sheet, which was 2% agarose M (Amersham Pharmacia Biotech) in DB without Ca$^{2+}$ or Mg$^{2+}$ dissolved in a microwave oven and solidified between two glass slides spaced with coverslips. Excess fluid was removed with filter paper. After incubation in a moist chamber for at least 20 min, the coverslip was set in an Attofluor™ cell chamber (ThermoFisher Scientific) and observed under TIRFM. For inhibitor treatment, DB containing 5–10 µM latrunculin A (Sigma) and 40 µM LY294002 (Cayman Chemical) was used for the cell suspension, and a 1 cm$^2$ agarose sheet soaked with DB containing inhibitors for at least 30 min was used for the overlay. Single PTEN-Halo molecules labeled with TMR were observed under an objective type TIRFM constructed on an inverted fluorescence microscope (IX70, Olympus)[57]. The images were acquired at 30 frames per sec. For the quantification of the membrane-association frequency, single-molecule imaging and EPI fluorescent imaging were successively performed in the same cells under different incident angles of the excitation light achieved by tilting the mirror on the light path.

**Subcellular localization biochemical analysis.** 1200 µL suspension of the starved cells at 1 × 10$^7$ cells per mL was mixed with 300 µL DB containing 50 µM latrunculin A in a 15-mL tube and incubated with shaking at 150 r.p.m. for 30 min. This step was duplicated. During the last 5 min, 20 µL DB containing 4.5 mM LY294002 was added to one sample. After centrifugation at 500xg for 2 min, the supernatant was discarded, and the cells were re-suspended in 150 µL DB. 100 µL cell suspension was immediately mixed with 100 µL basal buffer (20 mM Tris–HCl, 2 mM MgSO$_4$, pH 8.0) in a pre-chilled syringe (1 mL, Terumo), and the cell lysate passed through two layers of membrane (Whatman Nuclepore track-etched membrane, pore size 5.0 µm, Sigma-Aldrich) was collected in a tube on ice. 108 µL of the lysate was transferred to a new tube on ice and centrifuged at 16000 × *g* for 1 min at 4 °C. A concentration of 36 µL of the supernatant was transferred to a new tube containing 12 µL 4× sample buffer (β-mercaptoethanol-added, Wako). The residual supernatant was discarded, and the pellet was suspended in 48 µL 1× sample buffer. The supernatant and pellet fractions were boiled for 5 min and stored at −30 °C. 10 µL of the samples was subjected to SDS-PAGE in a precast gel (SuperSep Ace 5–20%, Wako) and blotted onto a PVDF membrane (Immobilon-P, Millipore). For the Western blot analysis of PTEN-Halo, the membrane was treated with 5% (w/v) skim milk/TBS-T (Tris-buffered saline containing 0.05% Tween-20) for 1 h at room temperature (RT), washed with TBS-T, and reacted with anti-HaloTag mouse monoclonal antibody (G921A, Promega) diluted with TBS-T at 1:1000 for 1 h at RT. The membrane was washed with TBS-T and reacted with horseradish peroxidase (HRP)-conjugated anti-mouse IgG (NA931, GE Healthcare) diluted with TBS-T at 1:50000 for 1 h at RT. After washing with TBS-T, signals were detected by using ECL Prime (GE Healthcare). For the analysis of PH$_{PKB}$-eGFP, the antibodies on the membrane were stripped by washing with TBS-T, and the membrane was treated with 5% skim milk/TBS-T for 1 h at RT and reprobed with anti-GFP pAb-HRP-DirecT (598-7, MBL) diluted with 1% skim milk/TBS at 1:2000 for 1 h at RT.

**Cell migration analysis.** The *x*- and *y*-coordinates of a cell in a movie were determined semi-automatically using laboratory-made software, and a migration trajectory for 30 min was made consisting of 361 positions at an interval of 5 s[58] (Supplementary Figs. 1b and 2ab). Displacement of the cell was measured every 5 s and multiplied by 12 to calculate the instantaneous velocity of the cell migration (µm min$^{-1}$). Measurement of the instantaneous velocity from more than 50 cells, each of which having 360 values, yielded a mean and five-number summary that included a minimum, lower quartile, median, upper quartile and maximum, as shown in the box-whisker plot (Fig. 1f; Supplementary Table 1). To quantify the chemotactic index, the trajectory was relocated so that the position of the cell at *t* = 0 was located on the origin and the position of the micropipette tip releasing cAMP was located on the *y*-axis (*x* = 0, *y* > 0) (Supplementary Fig. 2a, b). A mean of the y-component of the 5-sec displacement was divided by the mean of the 5-sec displacement in two dimensions (2D), yielding a chemotactic index of the cell (Supplementary Table 1). The exact value of *n*, which is the number of cells analyzed, can be found in Supplementary Table 1.

**Subcellular localization analysis**. Image analysis of the spatiotemporal dynamics of PTEN and PIP3 enrichment on the cell membrane was performed based on the kymographs obtained using a macro in Image-Pro Plus[15] (Media Cybernetics) (Fig. 1h, k; Supplementary Fig. 1c-e). The fluorescence intensities of PTEN-Halo-TMR and PH$_{PKB}$-eGFP were measured along the periphery of a circular ROI set on a cell image. The diameter of the ROI was kept constant throughout a stack of 361 images acquired at 5-sec intervals for 30 min. All cells that showed significant expression of both PTEN-Halo-TMR and PH$_{PKB}$-eGFP were used for the measurement, and multiple stacks obtained by the experiments performed on different days were analyzed. Calculations of the auto-correlation, oscillation period and averaged intensity distribution along the cell periphery were performed using MATLAB (MathWorks) (Fig. 1m–o; Supplementary Fig. 2d). Data are represented as means ± SDs. The exact value of $n$, which is the number of cells analyzed, can be found in Results and the figure legends.

To analyze the mutually inhibitory relationship between PTEN and PIP3 levels on the cell membrane, the fluorescence intensities of PTEN-Halo-TMR and PH$_{PKB}$-eGFP were measured using the plot profile command against the circular line set on the periphery of a cell image in ImageJ (Figs. 2; Supplementary Fig. 3b–d, 4a, c, d, f). A typical periphery of a cell observed under laser confocal microscopy was approximately 30 μm, which corresponds to 300 pixels in the 1024 × 1024 pixel image acquired using a 60× objective lens and 2× optical zoom by Nikon A1. Measurements in the TMR and GFP channels within the same microscopic ROI yielded approximately 300 data sets of TMR and GFP intensities along the periphery. The TMR and GFP intensities were normalized to the mean TMR and GFP intensities measured in the cytoplasm, respectively. The data sets obtained from 5 representative cells were displayed in a heat scatter plot, in which a single data set of the normalized TMR and GFP intensities is represented by a single point plotted in the PTEN-PIP3 plane. The same cells were analyzed throughout the manipulation of the LY294002 concentration. A histogram of the plotted points in the PTEN-PIP3 plane was used to calculate the probability density distribution in 2D, which is shown as a heat map. Two heat maps at 1 and 40 μM LY294002 were respectively fitted to a 2D Gaussian distribution to generate isodensity contours, which were used to estimate the border between the two distributions (Supplementary Fig. 3c). The heat scatter plots and heat maps were made using MATLAB. The exact value of $n$, which is the number of cells analyzed, can be found in "Results" section and the figure legends.

To analyze temporal changes in the subcellular localization upon the addition/dilution of LY294002 or cAMP, fluorescence intensities on the cell membrane or in the cytoplasm were measured and normalized to a mean fluorescent intensity measured before the treatment (Fig. 3c, d; Supplementary Figs. 2e, 3a). The data are represented as means ± SDs. The exact value of $n$, which is the number of cells analyzed, can be found in the figure legends.

**Single-molecule tracking**. A trajectory of a single PTEN-Halo-TMR molecule undergoing lateral diffusion on the cell membrane was obtained semi-automatically by using laboratory-made software[35,59]. The $x$- and $y$-coordinates of the molecule were determined by fitting the fluorescence intensity distribution to a 2D Gaussian distribution every 33 msec. All the molecules detected in the movie were tracked when possible, and those detectable at the onset and end of the movie were discarded from the analysis. An ensemble of $N$ trajectories sampled under the same experimental conditions was designated as $(x_i(t), y_i(t))$, where $i = 1, 2, \dots N$ and $t = 0$ means the time when the $i$-th molecule appeared on the membrane. The exact values of $N$ and $n$, which are the number of trajectories and cells analyzed, respectively, can be found in Table 1.

**Single-molecule lifetime analysis**. The number of molecules was counted at $t$ ($t = 0, 0.33, 0.66, 0.10, \dots$) and defined as $N$ at $t = 0$. The number divided by $N$ was plotted along $t$ to make a dissociation curve (Fig. 4c). Data in Fig. 4c are represented as means. The dissociation curve was fitted to an exponential function described as,

$$P_J^k(t) = \sum_{j=1}^{J} q_j^k e^{-\left(k_j + k_b\right)t},\qquad(1)$$

where $k_j$ denotes the $j$-th rate constant of the decay and $k_b$ denotes a rate constant of TMR photo-bleaching. Because the dissociation curve exhibited a faster decay than the photo-bleaching, it reflects the membrane-dissociation kinetics of PTEN. The dissociation curve before $t = 7$ s was well fitted to equation (1) with $J = 3$ and was consistent with the number of membrane-binding states[25] (Table 1). $k_b$ was 0.1 s$^{-1}$ and estimated by fitting the dissociation curve of TMR molecules immobilized on a coverslip by nonspecific adsorption. These molecules were visualized and tracked under the same conditions[35]. The lifetime of membrane binding was quantified as,

$$\tau = \frac{1}{q_1/k_1 + q_2/k_2 + q_3/k_3}\left(\frac{q_1}{k_1^2} + \frac{q_2}{k_2^2} + \frac{q_3}{k_3^2}\right).\qquad(2)$$

**Single-molecule diffusion analysis**. The displacement of a molecule in every 33-msec time interval of the trajectory was measured irrespective of $t$. The displacement, $\Delta r$, in the time interval, $\Delta t$, follows a probability density function (PDF) described as,

$$P_J^D(\Delta r, \Delta t) = \sum_{j=1}^{J} q_j^D \frac{\Delta r}{2D_j \Delta t + 2\varepsilon^2} e^{\frac{-\Delta r^2}{4D_j \Delta t + 4\varepsilon^2}}, \text{ where } \sum_{j=1}^{J} q_j^D = 1. \qquad(3)$$

$D_j$ and $q_j^D$ denote the diffusion coefficient and fraction of the $j$-th component ($j = 1, 2, \dots J$), respectively. $\varepsilon$ denotes a standard deviation of a measurement error, which was estimated to be less than 23 nm by a linear regression of mean squared displacements with lag time $\tau = 0.033$ or $0.066$ s using the equation[35] $MSD(\tau) = 4D\tau + 4\varepsilon^2$. $D_j$ and $q_j^D$ with variable $J$ ($J = 1,2,3,4$) were estimated by maximum likelihood estimation (MLE) using $M$ samples of $\Delta r_m$ ($m = 1, 2, \dots M$) measured in $N$ trajectories. A model assuming three components ($J = 3$) was most likely based on Akaike's Information Criterion (AIC)[35] (Fig. 5a; Table 1). AIC of the model assuming $J$ components, AIC$_J$, is described as,

$$\text{AIC}_J = -2L_J\left(\hat{\theta}_J\right) + (2J - 1)\log(\log M),\qquad(4)$$

where $\theta_J = (D_j, q_j^D)$ ($j = 1, 2, \dots J$) denotes a vector of the parameters to be estimated, and $L_J$ denotes a log likelihood written as

$$L_J(\theta_J) = \sum_{m=1}^{M} \log P_J^D\left(\Delta r_m | \theta_J\right).\qquad(5)$$

The number of trajectories, $N$, is suggested to be sufficient to estimate the diffusion coefficients[34]. The effect of cAMP stimulation on the diffusion coefficients was analyzed by the same method as above using the trajectories obtained from the cells at the indicated time after the stimulation (Fig. 5d).

**Single-molecule lifetime-diffusion analysis**. Rate constants of the state transition and membrane dissociation were estimated according to the lifetime-diffusion analysis method proposed previously[25,35]. Briefly, using the displacement of molecules between $t$ and $t + 0.033$ s, the fraction of the $j$-th component ($j = 1, 2, 3$) as a function of $t$, $q_j^D(t)$, was obtained by MLE. Diffusion coefficients were fixed to the values estimated in the diffusion analysis described above (Table 1). For convenience, the molecules adopting either the primary or weakly binding states were regarded as adopting the transiently binding state, and $q_2^D(t)$ and $q_3^D(t)$ for the components with moderate and fastest diffusion coefficients, respectively, were summed and renamed as $q_{2'}^D(t)$ (Fig. 5b). The number of molecules adopting the stably or transiently binding states normalized to $N$ follows $P_1(t)$ or $P_{2'}(t)$, which are described respectively as,

$$\begin{aligned} dP_1(t)/dt &= -(k_{off1} + k_{12'})P_1(t) + k_{2'1}P_{2'}(t),\\ dP_{2'}(t)/dt &= -(k_{off2'} + k_{2'1})P_{2'}(t) + k_{12'}P_1(t). \end{aligned}\qquad(6)$$

$k_{12'}$ and $k_{2'1}$ denote the rate constant of the state transition from the stably to transiently binding state and from the transiently to stably binding state, respectively, and $k_{off1}$ and $k_{off2'}$ denote the rate constants of membrane dissociation from the stably and transiently binding states, respectively. Thus, the number of total molecules bound to the cell membrane normalized to $N$ is described as,

$$\begin{aligned} dP(t)/dt &= d(P_1(t) + P_{2'}(t))/dt\\ &= -k_{off1}P_1(t) - k_{off2'}P_{2'}(t). \end{aligned}\qquad(7)$$

$q_1^D(t)$, $q_{2'}^D(t)$ and the dissociation curve obtained in the lifetime analysis described above (Fig. 4c) were respectively fitted to $P_1(t)/(P_1(t) + P_{2'}(t))$, $P_{2'}(t)/(P_1(t) + P_{2'}(t))$ and $P(t)$, which were obtained by solving the simultaneous differential equation on MatLab assuming steady state and with initial probabilities at $t = 0$ of $P_1(0) = a$ and $P_{2'}(0) = 1-a$. Data before $t = 2$ s was used for the fitting by the least-square method with the dissociation curve 3-fold weighted compared to $q_1^D(t)$ and $q_{2'}^D(t)$ since the dissociation curve exhibited exponential decay (Fig. 5b, c).

**Statistics**. Sample sizes were determined based on experience and related literature. All the experiments were repeated at least twice with replicates of at least 2. No criteria of data inclusion or exclusion were used unless otherwise noted. The data were collected from randomly chosen cells. No blinding was performed. Statistical analysis was carried out as described above. The data are represented as either means ± SD, SE or means, which are indicated in figure legends. Statistical comparisons were performed with the two-tailed Welch's $t$ test, likelihood-ratio test or regression slope test between data groups.

**Data availability**
The data supporting the findings of this manuscript are available from the corresponding author upon reasonable request. The custom-written code for statistical

analysis of single-molecule trajectories can be found on github ([https://github.com/SatomiMatsuoka/EstDstates](https://github.com/SatomiMatsuoka/EstDstates)). All the other codes are available from the corresponding author upon reasonable request.

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

## Acknowledgements

A dual expression vector, pDM181, was a gift from P.J.M. Van Haastert. Genes coding for RasG$_{Q61L}$, RasC$_{Q62L}$ and Rap1$_{G12V}$ were gifts from Y. Kamimura. A plasmid for GFP-Nodulin expression was a gift from P.N. Devreotes. The authors would like to thank S. Fukushima for daily discussions and P. Karagiannis for critical reading of the manuscript. This work was supported by AMED-CREST (JP17gm0910001) from Japan Agency for Medical Research and Development, AMED and JSPS KAKENHI Grant Number JP25871120.

## Author contributions

S.M. and M.U. conceived and designed the research and wrote the manuscript. S.M. conducted all the experiments and analyses.

## Additional information

**Competing interests:** The authors declare no competing interests.

