## [Peer Review File · Nature Communications]

Reviewers' Comments:

Reviewer #1:

Remarks to the Author:

The main novelty in this manuscript is the conclusion that high PIP3 levels can counter-intuitively exclude the PIP3-phosphatase PTEN from the plasma membrane and that this contributes to a bi-stable system important for defining mutually exclusive regions of PIP3 and PTEN at the leading and trailing edges, respectively, of chemotaxing cells. This would be a significant advance on the now extensive published literature, both from this group and others, on the role of PI3K signalling during chemotaxis.

The expts are technically demanding and often difficult to present in a manner that accounts for statistical variability in the processes being studied. However, on balance, I thought the experimental approaches were cleverly designed and complemented each other and I was convinced by the key findings. The manuscript will be a challenge to read however for those outside of this field.

Specific comments that should be addressed:

1. Given the accepted importance of PIP2 in regulating PTEN membrane localisation, the levels of PIP2 should be measured under conditions of PIP3 manipulation (ie. myr-PI3K, Ras-CA, LY29004) to show that they are not affected as well (eg via indirect effects on PIP2 consumption by PLC or PI3Ks).
2. HsPTEN is convincingly shown to lack PIP3-sensitive stable binding to the membrane and this impacts on PH-domain/PTEN boundary features, but this protein seems to rescue most of the chemotaxis defect; doesn't this imply that the PIP3 regulation of PTEN-membrane binding is really not that important for motility/chemotaxis?
3. I would like to see a discussion of the current disagreement in the field over the central role for PI3K signalling in Dicty chemotaxis – what is the authors take on reports that Dicty mutants that lack PI3Ks (used in this study) can chemotax efficiently?
4. In the Discussion, the authors imply that the main effect of cAMP signalling is to re-orientate excitable PIP3 patches in the direction of the gradient (by locally inhibiting the stable binding site for PTEN - could this be a PLC-mediated reduction in PIP2?), but recent biochemical expts clearly show cAMP stimulates total PIP3 production, even in the absence of PTEN (ie a large nett increase in PIP3 molecules; doi: 10.15252/embj.201488677), pointing to the importance of cAMP-stimulated PI3K activity?
5. I would like to see images from both WT and PTEN-null cells presented alongside the existing images in Fig 1A so the reader can judge the impact of PTEN deletion on the extent of PH-domain accumulation and the effect of colour overlap when TMR-Halo staining is present or absent.
6. It would be easier to independently evaluate the distribution of PTEN and PH-domain in some of the images if the eGFP and TMR-Halo images were separated, and not always shown as superimposed eg Fig 1A.
7. The supplementary section should include TMR-staining of a PTEN-null cell to evaluate the extent of background fluorescence.
8. The use of caffeine to inhibit the cAMP relay should be explained more clearly
9. Supplementary Figure 2C – why were the DdPTEN-G129E-Halo expressing PTEN-null cells treated with 40micromolar LY294002?

10. The Legend to supplementary Fig 5 needs to be corrected

11. Presumably there must be other PIP3 phosphatases present in Dicty, but this is not discussed?

Reviewer #2:

Remarks to the Author:

This is an impressive manuscript addressing a fundamental problem in signaling biology. The manuscript investigates the mechanism by which PTEN is excluded from membrane regions where PI3K and PIP3 production are activated, thereby enabling the creation of localized PIP3 domains, such as at the leading edge membrane of polarized chemotaxing cells.

The manuscript provides extensive and clear evidence of a bistable relationship between PIP3 density and PTEN density on the membrane surface, such that high PIP3 density blocks stable PTEN binding while low PIP3 density allows stable PTEN binding. Multiple approaches are utilized to probe the system in complementary, independent directions, yielding large data sets that are carefully analyzed and described.

Overall, this is an important contribution. My comments are:

1. (Mechanistic Model, Fig 5E) The model is well supported by the data. An interesting question raised by the model is whether the "stable state" of PTEN bound to the unidentified stabilizer protein is able to catalyze multiple PIP3 hydrolysis events, while the "unstable states" not bound to the stabilize protein catalyze zero or one turnovers. The proposal that PIP3 can bind to the active site of the unstable state and thereby inhibit association of PTEN with the stabilizer protein would suggest this state could be a single turnover state. In vivo (or in vitro if necessary) single molecule analysis may well be able to detect the single turnovers of the unstable state. Similarly, such analysis may well be able to detect the number of turnovers of the stable state. Such questions are likely outside the scope of the current manuscript, however.

2. (Throughout) The language of the manuscript can lead to the mistaken impression that the "binding site" inhibited by PIP3 is on PTEN, rather than on a different stabilizer protein that associates with PTEN in the stably bound state. This leads to some confusion for the reader that is finally clarified by the model in the last figure, Fig 5E. It would be helpful to more clearly describe the model earlier in the MS, or perhaps move Fig 5E to the Introduction.

3. (Pg 13, lines 237-243) In a key section of the Results describing the fractions of the population in states exhibiting different binding and diffusion parameters, the language of the description is potentially misleading.

Reviewer #3:

Remarks to the Author:

The mutual exclusion of PIP3 and PTEN on the membrane of migrating cells has been carefully studied and analyzed in this paper. Although this exclusion is well documented, it is true that a proper mechanism for this regulation is lacking. Although some papers claim the PTEN binds to PI(4,5)P2 owing to the existence of a PH domain, this claim is vague at best. The breakdown of PIP3 by PTEN is again very well documented, but the decrease of PTEN at PIP3 enriched sites is unexplained. This paper provides a good mechanistic basis by which this mutual inhibition may occur. The suppression of stable binding sites and increased membrane dissociation of PTEN by PIP3 provide two reasonable regulating mechanisms for the missing arm of this mutual inhibition.

This regulating mechanism which is essentially outlined in Figure 4, 5 and 6 is novel and interesting. However, other than this mechanism, the cell migration context of the paper is somewhat lacking. That is, the title “Mutual inhibition generates bistability for cell motility” is misleading as the paper is more about the proof and mechanism for mutual inhibition. The “bistability for cell motility” argument is rather confusing and ill-explained for reasons stated below.

The authors provide illustrations of how PIP3 and PTEN exist in bistable states, where there are two major peaks – one at [lowPIP3,highPTEN] or at [highPIP3,lowPTEN]. This is their major argument for bistability. However, how this bistability contributes to cell motion is not addressed. It is well established that the system that controls the signaling apparatus is an excitable network – and the authors agree to that fact. They mention that the bistability of PIP3 manifests in the presence of a gradient, explaining the continuous domain generation. Does this mean that the cell gets rid of excitability and adopts bistability in the presence of a gradient? Major work by other groups such as Gerisch, Devreotes, etc would disagree and their work shows that the excitable waves still remain in the presence of a gradient. Hence, in order to name the paper “Mutual inhibition generates bistability for cell motility” – the context of bistability and how it fits into the excitable arguments for cell motion has to be further explained. This paper would be better named as: “Mutual inhibition mechanism for the regulation of PTEN and PIP3”, as the “bistability for cell motility” argument can be misleading in the present state.

For the context of cell migration, my major suggestion would be to incorporate a mathematical model of this bistability and to show how this can also account for the excitable behavior. The same excitable system can in fact generate both monostable and bistable states. The two mechanisms of PIP3 regulation (phosphatase-dependent and –independent) can be the two control parameters in this model.

Other concerns/suggestions about the paper are outlined below:

Result 1 (corresponding to figure 1): The fig1 d-g middle panels do not look like the pten-null kymographs from the supplemental, although the phenotype is similar. Is there a problem with the color adjustment/quantification or is this because of the Latrunculin? Similarly, although the paper claims that PIP3 is “increased uniformly” in the G129E mutant, the level of PIP3 does not seem to be different in fig1 E and F middle panels when compared with the DdPTEN left panel. On the other hand however, the plots in fig1 I (middle panel), agree with the PIP3 increase for the same mutant. This is somewhat confusing. The amount of PIP3 in the cell, after these mutants are introduced, should be quantified and reported in the first figure.

Later in the paper, the authors show that increased PIP3 results in increased dissociation of PTEN from the membrane. Why do we not see that in the G129E mutant? As the phosphatase activity is lacking, the amount of PIP3 should be reasonably high. Should the high PIP3 not affect the membrane localization of the PTEN (as suggested in Figure 4 through increased membrane dissociation), and cause the cell to recover?

Most importantly, the amount of PIP3 in each of these three situations should be quantified and reported before these results are introduced.

Result 2 (corresponding to figure 2): It is here that a bistable model should be introduced to make this idea clearer.

Why was the magenta line removed as a reference in the middle panel of B? The old position of the line from the top panel in B should be overlaid in the middle panel so as to demonstrate how co-localization occurs and the bistability is lost.

Result 3 (corresponding to figure 3): In the bottom panel of A, it is claimed that the exclusion of

DdPTEN does not occur. Although it does look that way from the figure, this needs quantification for better illustration, and for comparison with the +LY result as this phenotype seems somewhat weaker. Perhaps something like membrane to cytosol fluorescence intensity ratio is needed. p values are needed in panel D. (p-values are, in general, lacking from all plots in this paper, although the methods claim that the standard t-test was done).

Result 4 (corresponding to figure 4): Last paragraph: The scattered plot of the number of molecules between PTEN-Halo-TMR and the membrane. Should there not be a figure reference here?

Result 5 (corresponding to figure 5): Error-bars and p-values are missing in plots A and D.

Reviewers' comments:

Reviewer #1 (Remarks to the Author):

The main novelty in this manuscript is the conclusion that high PIP3 levels can counter-intuitively exclude the PIP3-phosphatase PTEN from the plasma membrane and that this contributes to a bi-stable system important for defining mutually exclusive regions of PIP3 and PTEN at the leading and trailing edges, respectively, of chemotaxing cells. This would be a significant advance on the now extensive published literature, both from this group and others, on the role of PI3K signalling during chemotaxis.

The expts are technically demanding and often difficult to present in a manner that accounts for statistical variability in the processes being studied. However, on balance, I thought the experimental approaches were cleverly designed and complemented each other and I was convinced by the key findings. The manuscript will be a challenge to read however for those outside of this field.

Specific comments that should be addressed:

1. Given the accepted importance of PIP2 in regulating PTEN membrane localisation, the levels of PIP2 should be measured under conditions of PIP3 manipulation (ie. myr-PI3K, Ras-CA, LY29004) to show that they are not affected as well (eg via indirect effects on PIP2 consumption by PLC or PI3Ks).
2. HsPTEN is convincingly shown to lack PIP3-sensitive stable binding to the membrane and this impacts on PH-domain/PTEN boundary features, but this protein seems to rescue most of the chemotaxis defect; doesn't this imply that the PIP3 regulation of PTEN-membrane binding is really not that important for motility/chemotaxis?

3. I would like to see a discussion of the current disagreement in the field over the central role for PI3K signalling in Dicty chemotaxis – what is the authors take on reports that Dicty mutants that lack PI3Ks (used in this study) can chemotax efficiently?
4. In the Discussion, the authors imply that the main effect of cAMP signalling is to re-orientate excitable PIP3 patches in the direction of the gradient (by locally inhibiting the stable binding site for PTEN - could this be a PLC-mediated reduction in PIP2?), but recent biochemical expts clearly show cAMP stimulates total PIP3 production, even in the absence of PTEN (ie a large nett increase in PIP3 molecules; doi: 10.15252/emj.201488677), pointing to the importance of cAMP-stimulated PI3K activity?
5. I would like to see images from both WT and PTEN-null cells presented alongside the existing images in Fig 1A so the reader can judge the impact of PTEN deletion on the extent of PH-domain accumulation and the effect of colour overlap when TMR-Halo staining is present or absent.
6. It would be easier to independently evaluate the distribution of PTEN and PH-domain in some of the images if the eGFP and TMR-Halo images were separated, and not always shown as superimposed eg Fig 1A.
7. The supplementary section should include TMR-staining of a PTEN-null cell to evaluate the extent of background fluorescence.
8. The use of caffeine to inhibit the cAMP relay should be explained more clearly
9. Supplementary Figure 2C – why were the DdPTEN-G129E-Halo expressing PTEN-null cells treated with 40micromolar LY294002?
10. The Legend to supplementary Fig 5 needs to be corrected
11. Presumably there must be other PIP3 phosphatases present in Dicty, but this is not discussed?

Reviewer #2 (Remarks to the Author):

This is an impressive manuscript addressing a fundamental problem in signaling biology. The manuscript investigates the mechanism by which PTEN is excluded from membrane regions where PI3K and PIP3 production are activated, thereby enabling the creation of localized PIP3 domains, such as at the leading edge membrane of polarized chemotaxing cells.

The manuscript provides extensive and clear evidence of a bistable relationship between PIP3 density and PTEN density on the membrane surface, such that high PIP3 density blocks stable PTEN binding while low PIP3 density allows stable PTEN binding. Multiple approaches are utilized to probe the system in complementary, independent directions, yielding large data sets that are carefully analyzed and described.

Overall, this is an important contribution. My comments are:

1. (Mechanistic Model, Fig 5E) The model is well supported by the data. An interesting question raised by the model is whether the "stable state" of PTEN bound to the unidentified stabilizer protein is able to catalyze multiple PIP3 hydrolysis events, while the "unstable states" not bound to the stabilize protein catalyze zero or one turnovers. The proposal that PIP3 can bind to the active site of the unstable state and thereby inhibit association of PTEN with the stabilizer protein would suggest this state could be a single turnover state. In vivo (or in vitro if necessary) single molecule analysis may well be able to detect the single turnovers of the unstable state. Similarly, such analysis may well be able to detect the number of turnovers of the stable state. Such questions are likely outside the scope of the current manuscript, however.

2. (Throughout) The language of the manuscript can lead to the mistaken impression that the "binding site" inhibited by PIP3 is on PTEN, rather than on a different stabilizer protein that associates with PTEN in the stably bound state. This leads to some confusion for the reader that is finally clarified by the model in the last figure, Fig 5E. It would be helpful to more clearly describe the model earlier in the MS, or perhaps move Fig 5E to the Introduction.

3. (Pg 13, lines 237-243) In a key section of the Results describing the fractions of the population in states exhibiting different binding and diffusion parameters, the language of the description is potentially misleading.

Reviewer #3 (Remarks to the Author):

The mutual exclusion of PIP3 and PTEN on the membrane of migrating cells has been carefully studied and analyzed in this paper. Although this exclusion is well documented, it is true that a proper mechanism for this regulation is lacking. Although some papers claim the PTEN binds to PI(4,5)P2 owing to the existence of a PH domain, this claim is vague at best. The breakdown of PIP3 by PTEN is again very well documented, but the decrease of PTEN at PIP3 enriched sites is unexplained. This paper provides a good mechanistic basis by which this mutual inhibition may occur. The suppression of stable binding sites and increased membrane dissociation of PTEN by PIP3 provide two reasonable regulating mechanisms for the missing arm of this mutual inhibition.

This regulating mechanism which is essentially outlined in Figure 4, 5 and 6 is novel and interesting. However, other than this mechanism, the cell migration context of the paper is somewhat lacking. That is, the title “Mutual inhibition generates bistability for cell motility” is misleading as the paper is more about the proof and mechanism for mutual inhibition. The “bistability for cell motility” argument is rather confusing and ill-explained for reasons stated below.

The authors provide illustrations of how PIP3 and PTEN exist in bistable states, where there are two major peaks – one at [lowPIP3,highPTEN] or at [highPIP3,lowPTEN]. This is their major argument for bistability. However, how this bistability contributes to cell motion is not addressed. It is well established that the system that controls the signaling apparatus is an excitable network – and the authors agree to that fact. They mention that the bistability of PIP3 manifests in the presence of a gradient, explaining the continuous domain generation. Does this mean that the cell gets rid of excitability and adopts bistability in the presence of a gradient? Major work by other groups such as Gerisch, Devreotes, etc would disagree and their work shows that the excitable waves still remain in the presence of a gradient. Hence, in order to name the paper “Mutual inhibition generates bistability for cell motility” – the context of bistability and how it fits into the excitable arguments for cell motion has to be further explained. This paper would be better named as: “Mutual inhibition mechanism for the regulation of PTEN

and PIP3”, as the “bistability for cell motility” argument can be misleading in the present state.

For the context of cell migration, my major suggestion would be to incorporate a mathematical model of this bistability and to show how this can also account for the excitable behavior. The same excitable system can in fact generate both monostable and bistable states. The two mechanisms of PIP3 regulation (phosphatase-dependent and – independent) can be the two control parameters in this model.

Other concerns/suggestions about the paper are outlined below:

Result 1 (corresponding to figure 1): The fig1 d-g middle panels do not look like the pten-null kymographs from the supplemental, although the phenotype is similar. Is there a problem with the color adjustment/quantification or is this because of the Latrunculin? Similarly, although the paper claims that PIP3 is “increased uniformly” in the G129E mutant, the level of PIP3 does not seem to be different in fig1 E and F middle panels when compared with the DdPTEN left panel. On the other hand however, the plots in fig1 I (middle panel), agree with the PIP3 increase for the same mutant. This is somewhat confusing. The amount of PIP3 in the cell, after these mutants are introduced, should be quantified and reported in the first figure.

Later in the paper, the authors show that increased PIP3 results in increased dissociation of PTEN from the membrane. Why do we not see that in the G129E mutant? As the phosphatase activity is lacking, the amount of PIP3 should be reasonably high. Should the high PIP3 not affect the membrane localization of the PTEN (as suggested in Figure 4 through increased membrane dissociation), and cause the cell to recover?

Most importantly, the amount of PIP3 in each of these three situations should be quantified and reported before these results are introduced.

Result 2 (corresponding to figure 2): It is here that a bistable model should be introduced to make this idea clearer.

Why was the magenta line removed as a reference in the middle panel of B? The old position of the line from the top panel in B should be overlaid in the middle panel so as to demonstrate how co-localization occurs and the bistability is lost.

Result 3 (corresponding to figure 3): In the bottom panel of A, it is claimed that the exclusion of DdPTEN does not occur. Although it does look that way from the figure, this needs quantification for better illustration, and for comparison with the +LY result as this phenotype seems somewhat weaker. Perhaps something like membrane to cytosol fluorescence intensity ratio is needed.

p values are needed in panel D. (p-values are, in general, lacking from all plots in this paper, although the methods claim that the standard t-test was done).

Result 4 (corresponding to figure 4): Last paragraph: The scattered plot of the number of molecules between PTEN-Halo-TMR and the membrane. Should there not be a figure reference here?

Result 5 (corresponding to figure 5): Error-bars and p-values are missing in plots A and D.

Response to the reviewers' comments:

Our response to Reviewer #1:

We are grateful to reviewer #1 for the important suggestions that have helped improve our manuscript. As indicated in the following responses, we have taken all the comments and suggestions into account in the revised version of our manuscript. We put the comments made by the reviewer in ***bold italics*** with our responses below. Our changes in the revised manuscript are marked with highlighted **green-colored font**.

1) Given the accepted importance of PIP2 in regulating PTEN membrane localisation, the levels of PIP2 should be measured under conditions of PIP3 manipulation (ie. myr-PI3K, Ras-CA, LY29004) to show that they are not affected as well (eg via indirect effects on PIP2 consumption by PLC or PI3Ks).

We have included the results of PIP2 imaging and quantification in the revised manuscript. Briefly, PI(4,5)P2 detected with GFP-Nodulin, which is a specific fluorescent probe established in *Arabidopsis thaliana* and *Saccharomyces cerevisiae* [Ghosh et al., *Mol. Biol. Cell*, 2015], exhibited a slight decrease with increasing PI(3,4,5)P3 on the cell membrane (line 7, pp. 11; Supplementary Fig. 4a and b). The fluorescence intensities of the PIP2 and PIP3 probes, GFP-Nodulin and PH_{PKB}-RFP, respectively, exhibited negative correlation, showing that the sum of these two phosphoinositides was almost constant during the experiments. We confirmed that PIP2 positively regulated PTEN membrane localization at 40 μ M LY294002 (full inhibition of PI3K activity; lines 8-20, pp. 10; Fig. 2d, upper left). However, in the absence of LY294002, PIP3 production by membrane-tethered PI3K (myrPI3K2) caused a decrease in PTEN membrane localization in PIP2-dependent and -independent manners (Fig. 2d, upper right). As a result, given an arbitrary PIP2 level, the PTEN and PIP3 levels followed a bimodal distribution, indicating bistability (Fig. 2e). The extra decrease in PTEN membrane localization coupled with PIP3 production was clearly seen in migrating *D. discoideum* cells (lines 11-17, pp. 5; Fig. 1b). DdPTEN was almost always excluded from the anterior membrane where PIP3 was enriched, even if PIP2 was not

obviously reduced compared to the lateral and posterior membrane. These observations indicate that DdPTEN membrane localization is not solely dependent on PIP2 but affected negatively by PIP3. We believe that after inclusion of these data the manuscript has been improved significantly.

2) HsPTEN is convincingly shown to lack PIP3-sensitive stable binding to the membrane and this impacts on PH-domain/PTEN boundary features, but this protein seems to rescue most of the chemotaxis defect; doesn't this imply that the PIP3 regulation of PTEN-membrane binding is really not that important for motility/chemotaxis?

We apologize for the confusion in Fig.1. In order to demonstrate clearly the defect of HsPTEN-expressing cells, we have included statistical data of the pseudopod projections (lines 3-4, 7-8 and 12-13, pp. 6; Fig. 1e) and movies of DdPTEN-, DdPTEN_{G129E}- or HsPTEN-Halo-expressing *pten*-null cells under chemotaxis (line 14, pp. 6; Supplementary Movies 1-3) as well as a brief discussion (lines 12-15, pp. 17). We believe that these revisions have made the claim convincing that PTEN membrane localization contributes to directed cell migration by confining the PIP3-enriched domain and thus restricting the direction of pseudopod projection.

3) I would like to see a discussion of the current disagreement in the field over the central role for PI3K signalling in Dicty chemotaxis – what is the authors take on reports that Dicty mutants that lack PI3Ks (used in this study) can chemotax efficiently?

We have included a discussion on the role of PI3K signaling in cell motility in the revised manuscript (line 12, pp. 17 - line 9, pp. 18). Briefly, the fact that *D. discoideum* mutants that lack PI3K enzymatic activity still undergo chemotaxis [Hoeller and Kay, *Curr. Biol.*, 2007] indicates that the role of PI3K in chemotaxis can be replaced by other molecules involved in TorC2, sGC and PLA2 signaling, which are activated upon chemoattractant stimulation in parallel with PI3K signaling [Chen et al., *Dev. Cell*, 2007; Veltman et al., *J. Cell Biol.*, 2008]. These redundant signaling pathways are most

likely activated at the leading edge to induce asymmetric motility. On the other hand, a single gene knockout of PTEN causes a loss of cell motility with enhanced PIP3 accumulation and multiple lateral pseudopods as shown by previous studies [Iijima and Devreotes, *Cell*, 2002; Wessels et al., *J. Cell Sci.*, 2007] and the current manuscript. This finding indicates that PIP3 can trigger pseudopod formation and that the role of PTEN in cell motility is irreplaceable in the presence of PI3K activity. This manuscript provides evidence for the idea that PTEN confines the PIP3 patch through bistability, at least in part. Therefore, we think PI3K signaling may not be solely responsible for chemotaxis, but that it contributes to pseudopod formation with other signaling pathways.

4) In the Discussion, the authors imply that the main effect of cAMP signalling is to re-orientate excitable PIP3 patches in the direction of the gradient (by locally inhibiting the stable binding site for PTEN - could this be a PLC-mediated reduction in PIP2?), but recent biochemical expts clearly show cAMP stimulates total PIP3 production, even in the absence of PTEN (ie a large net increase in PIP3 molecules; doi: 10.15252/embj.201488677), pointing to the importance of cAMP-stimulated PI3K activity?

We agree with the reviewer about the importance of cAMP-stimulated PI3K activity in the re-orientation of the excitable PIP3 patches. In order to avoid misleading the readers, we have rewritten the phrase in the revised manuscript (line 2, pp. 21).

5) I would like to see images from both WT and PTEN-null cells presented alongside the existing images in Fig 1A so the reader can judge the impact of PTEN deletion on the extent of PH-domain accumulation and the effect of colour overlap when TMR-Halo staining is present or absent.

We have included images of PH_{PKB}-eGFP-expressing wild-type and *pten*-null cells taken under confocal fluorescence microscopy in the revised manuscript (lines 8-11, pp. 5; Fig. 1a).

6) It would be easier to independently evaluate the distribution of PTEN and PH-domain in some of the images if the eGFP and TMR-Halo images were separated, and not always shown as superimposed eg Fig 1A.

The eGFP and TMR channels of the images are separately shown in Fig. 1g and 1j in the revised manuscript.

7) The supplementary section should include TMR-staining of a PTEN-null cell to evaluate the extent of background fluorescence.

We have included negative control data in the revised manuscript (line 12-14, pp. 12; Supplementary Fig. 5a). The data demonstrate a negligible contribution of the background fluorescence, which may be caused by unspecific bindings of TMR-Halo ligands to the cell membrane and membrane proteins, to the single-molecule measurements or confocal observations.

8) The use of caffeine to inhibit the cAMP relay should be explained more clearly.

We have included an explanation for the use of caffeine with emphasis on its inhibitory effect on intercellular signaling (line 7, pp. 7).

9) Supplementary Figure 2C – why were the DdPTEN-G129E-Halo expressing PTEN-null cells treated with 40micromolar LY294002?

The treatment with LY294002 somehow improved the response of DdPTEN_{G129E}-Halo to the concentration gradient of cAMP in *pten*-null cells, although we currently have no idea of the mechanism. Since the response is reproducibly observed without LY294002 when stimulated with the concentration jump as shown in Supplementary Fig. 2e, this unknown mechanism for directional sensing may be involved in regulation of the assumptive binding site for PTEN. Previous studies have reported that defects in the cell

polarity and motility of *pten*-null cells can be restored by disruption of PKB signaling with LY294002 or gene knock-out of PKB or its downstream effector, PakA [Chen et al., *Mol. Biol. Cell*, 2003; Tang et al., *Mol. Biol. Cell*, 2012]. There may be some feedback regulation downstream of PKB upstream of the PTEN binding site, but the investigation of the mechanism is beyond the focus of this study.

10) The Legend to supplementary Fig 5 needs to be corrected.

We have corrected the legend of Supplementary Fig. 5.

11) Presumably there must be other PIP3 phosphatases present in Dicty, but this is not discussed?

We have included a brief discussion on the essential role of PTEN encoded by *ptenA*, which has been knocked out in this study, in *D. discoideum* cell motility (line 20, pp. 17 - line 4, pp. 18). Experimental data suggest that *ptenA* provides the major enzymatic activity of PIP3 3-phosphatase in chemotactically responding *D. discoideum* cells. Single gene knock-out of *ptenA* results in a constitutively high PIP3 level on the plasma membrane as shown in Fig. 1a and reported by Iijima and Devreotes [*Cell*, 2002]. The PIP3 level increased after cAMP stimulation hardly returns to the basal level within 45 sec, while it does in the presence of functional *ptenA* [Hoeller and Kay, *Curr. Biol.*, 2007]. Therefore, although other genes coding for PIP3 phosphatases are present in *D. discoideum* [Tang and Gomer, *Eukaryot. Cell*, 2008], it is unlikely that these gene products exert enough activity to keep cellular PIP3 levels low in chemotactically responding cells.

Our response to Reviewer #2:

We are grateful to reviewer #2 for the insightful comments that have helped improve our paper. We write the comments made by the reviewer in *bold italics* with our responses below. Our changes in the manuscript are marked with highlighted green-colored font.

1) (Mechanistic Model, Fig 5E) The model is well supported by the data. An interesting question raised by the model is whether the "stable state" of PTEN bound to the unidentified stabilizer protein is able to catalyze multiple PIP3 hydrolysis events, while the "unstable states" not bound to the stabilize protein catalyze zero or one turnovers. The proposal that PIP3 can bind to the active site of the unstable state and thereby inhibit association of PTEN with the stabilizer protein would suggest this state could be a single turnover state. In vivo (or in vitro if necessary) single molecule analysis may well be able to detect the single turnovers of the unstable state. Similarly, such analysis may well be able to detect the number of turnovers of the stable state. Such questions are likely outside the scope of the current manuscript, however.

We appreciate the valuable proposal from the reviewer. In fact, the relationship between the stability of membrane binding and the number of catalytic cycles is one of the most interesting problems we will tackle next. We actually tried to estimate the number from the single molecule trajectories, but currently have reached the conclusion that it is difficult to correlate enzymatic activity with the diffusion states. The reason is that phosphatase-dead PTEN (DdPTEN_{G129E} and DdPTEN_{C124S}, another mutant form of DdPTEN that is trapped in the substrate-bound state) exhibits the three diffusion states with the same diffusion coefficients as DdPTEN. In order not to mislead the readers, we have decided not to include any kind of pre-mature analyses on the number of dephosphorylation reactions. Instead, we have briefly described perspectives in the revised manuscript (lines 9-13, pp. 19).

2) (Throughout) The language of the manuscript can lead to the mistaken impression that the "binding site" inhibited by PIP3 is on PTEN, rather than on a different

stabilizer protein that associates with PTEN in the stably bound state. This leads to some confusion for the reader that is finally clarified by the model in the last figure, Fig 5E. It would be helpful to more clearly describe the model earlier in the MS, or perhaps move Fig 5E to the Introduction.

We have included a definition of the “binding site” in the revised manuscript to clarify that it describes some lipid/protein on the membrane that associates with PTEN (lines 18-19, pp. 13). Our model assumes that a specific binding site stabilizes PTEN in the stably bound state and that this binding site is inhibited by PIP3 through PTEN-dependent and –independent mechanisms. In order to avoid confusion, we have included the above description in the legend of Fig. 5e (lines 13-15, pp. 47).

3) (Pg 13, lines 237-243) In a key section of the Results describing the fractions of the population in states exhibiting different binding and diffusion parameters, the language of the description is potentially misleading.

The sentences have been rewritten in the revised manuscript (lines 4-10, pp. 14). We hope these revisions avoid confusion.

Our response to Reviewer #3:

We are grateful to reviewer #3 for the critical and informative comments that have helped improve our manuscript. As indicated in the responses that follow, we have addressed the comments in the revised version of our paper. We put the comments made by the reviewer in *bold italics* with our responses below. Our changes in the manuscript are marked with highlighted green-colored font.

This regulating mechanism which is essentially outlined in Figure 4, 5 and 6 is novel and interesting. However, other than this mechanism, the cell migration context of the paper is somewhat lacking. That is, the title “Mutual inhibition generates bistability for cell motility” is misleading as the paper is more about the proof and mechanism for mutual inhibition. The “bistability for cell motility” argument is rather confusing and ill-explained for reasons stated below.

The authors provide illustrations of how PIP3 and PTEN exist in bistable states, where there are two major peaks – one at [lowPIP3,highPTEN] or at [highPIP3,lowPTEN]. This is their major argument for bistability. However, how this bistability contributes to cell motion is not addressed. It is well established that the system that controls the signaling apparatus is an excitable network – and the authors agree to that fact. They mention that the bistability of PIP3 manifests in the presence of a gradient, explaining the continuous domain generation. Does this mean that the cell gets rid of excitability and adopts bistability in the presence of a gradient? Major work by other groups such as Gerisch, Devreotes, etc would disagree and their work shows that the excitable waves still remain in the presence of a gradient. Hence, in order to name the paper “Mutual inhibition generates bistability for cell motility” – the context of bistability and how it fits into the excitable arguments for cell motion has to be further explained. This paper would be better named as: “Mutual inhibition mechanism for the regulation of PTEN and PIP3”, as the “bistability for cell motility” argument can be misleading in the present state.

For the context of cell migration, my major suggestion would be to incorporate a mathematical model of this bistability and to show how this can also

account for the excitable behavior. The same excitable system can in fact generate both monostable and bistable states. The two mechanisms of PIP3 regulation (phosphatase-dependent and -independent) can be the two control parameters in this model.

We thank the reviewer for the valuable suggestions. We have included further evidence for the significance of bistability in the context of cell migration in Fig. 1 in the revised manuscript. Briefly, movies showing the chemotactic response of DdPTEN-, DdPTEN_{G129E}- and HsPTEN-Halo-expressing *pten*-null cells under concentration gradients of cAMP have been incorporated as Supplementary Movies 1-3. In the movie of the bistability-deficient HsPTEN cell (Supplementary Movie 3), PIP3 distributed on the cell membrane in a gradient manner from the front to back and induced pseudopod projection at the lateral and posterior sides. As a consequence, the directed migration was frequently halted, and thus the migration velocity was significantly reduced. Statistical data of the pseudopod projection direction have been also included (lines 3-4, 7-8 and 12-13, pp. 6; Fig. 1e). These results indicate PIP3-PTEN bistability contributes to directed cell migration through digitization of the PIP3 level at the border of the PIP3-enriched domain. We believe that these revisions provide an answer to the question on *how the bistability contributes to cell motion*, at least in part.

In order to respond to the reviewer's concern about *how the bistability fits into the excitable arguments for cell motion*, we have included new discussion in the revised manuscript (line 4, pp. 20 - line 3, pp. 21). Our model does not conflict with the recent arguments on excitability for cell motion. In prevailing models, the output of an excitable network is used as a local signal that governs cytoskeletal activities and thus cell motility [Arai et al., *PNAS*, 2010; Xiong et al., *PNAS*, 2010; Shibata et al., *Biophys. J.*, 2013]. For chemotaxis, a network for gradient sensing and the excitable network are connected in tandem to generate the local signal facing the source, as represented by the LEGI-STEN model [Tang et al., *Nat. Comm.*, 2014]. In our current model, the output of an excitable network is used as the input of a bistable network that generates the local signal. For chemotaxis, the output of a gradient sensing network is also used as the input of the bistable network independently of the excitable network. Thus, the local signal dynamics is coherent with the excitable network dynamics in the absence of the gradient, but the local signal is dependent on the gradient in its presence.

The local signal that governs cytoskeletal activities is thought to be a PIP3 patch, which we call a PIP3-enriched domain in the current manuscript [Arai et al., *PNAS*, 2010]. In the absence of the chemoattractant gradient, these patches exhibit excitable behaviors such as sporadic emergence, oscillation and traveling waves [Shibata et al., *J. Cell Sci.*, 2012; Miao et al., *Nat. Cell Biol.*, 2017]. On the other hand, several reports have found that these patches continuously face the chemoattractant source [Parent and Devreotes, *Science*, 1999; Janetopoulos et al, *PNAS*, 2004; Xu et al., *J Cell Biol.*, 2007; Wang et al., *PNAS*, 2013]. Currently, no model can account for all the spatiotemporal dynamics. A model that assumes tandem connection of the gradient-sensing and excitable networks cannot reconstitute the continuous patches, because the signal generated by any excitable network is essentially transient due to the refractory period of the network [Huang et al., *Nat. Cell Biol.*, 2013; Nishikawa et al., *Biophys. J.*, 2014]. When the excitable network is continuously activated to trigger excitation, it generates the same signal repeatedly and never generates a continuous signal [Tang et al., *Nat. Comm.*, 2014].

Our current model can explain all the above dynamics as follows. It assumes that Ras is activated by an excitable network, which is supported by evidence that RBD_{Raf1} exhibits traveling waves free from the activities of all 4 downstream signaling pathways (TorC2, sGC and PLA2 as well as PI3K) [Fukushima, Matsuoka and Ueda, <https://www.biorxiv.org/content/early/2018/08/09/356105>, 2018]. Owing to the direct interaction between active-form Ras and PI3K2 [Funamoto et al., *Cell*, 2002], we found synchronized traveling waves [paper under review: <https://www.biorxiv.org/content/early/2018/08/09/356105>]. PI3K activity serves as a control parameter for selecting the metastable states of the PIP3-PTEN bistable system (Fig. 2). Thus, RBD_{Raf1} and PIP3 become coupled to exhibit the same excitable behaviors in the absence of a chemoattractant, cAMP. Additionally, the cAMP gradient activates multiple PI3K subtypes [Takeda et al., *J. Biol. Chem.*, 2007] and inhibits PTEN by suppressing the stably binding state (Fig. 5d), which ensures the continuous and local state transition into the PIP3-enriched state without involving the excitable network.

We agree with the reviewer about the necessity of mathematical modeling. The unpublished paper mentioned in the previous paragraph describes a mathematical

model for the combined network of the excitable and bistable systems using reaction-diffusion equations. The model is constructed based on a substantial amount of experimental evidence obtained by fluorescence microscopy and statistical analysis. Therefore, we decided to separate the results into two manuscripts.

1) Result 1 (corresponding to figure 1): The fig1 d-g middle panels do not look like the pten-null kymographs from the supplemental, although the phenotype is similar. Is there a problem with the color adjustment/quantification or is this because of the Latrunculin? Similarly, although the paper claims that PIP3 is “increased uniformly” in the G129E mutant, the level of PIP3 does not seem to be different in fig1 E and F middle panels when compared with the DdPTEN left panel. On the other hand however, the plots in fig1 I (middle panel), agree with the PIP3 increase for the same mutant. This is somewhat confusing. The amount of PIP3 in the cell, after these mutants are introduced, should be quantified and reported in the first figure.

We apologize for the confusing images. We have replaced them with those that show higher expressions of both PH_{PKB}-eGFP and DdPTEN_{G129E}-Halo (Fig. 1g, middle panel). For better illustration, the kymographs in the original manuscript, which showed absolute fluorescence intensity measured at the cell periphery, have been replaced in the revised manuscript with those showing the membrane-cytoplasm ratio of the fluorescence intensity so that the degree of membrane localization can be compared among the three situations (Fig. 1h,k). In addition, we have included data showing average PIP3 levels in the three cell lines quantified by western blot detection of PH-eGFP with anti GFP antibody (lines 18-20, pp. 5; Fig. 1d).

2) Later in the paper, the authors show that increased PIP3 results in increased dissociation of PTEN from the membrane. Why do we not see that in the G129E mutant? As the phosphatase activity is lacking, the amount of PIP3 should be reasonably high. Should the high PIP3 not affect the membrane localization of the PTEN (as suggested in Figure 4 through increased membrane dissociation), and cause the cell to recover?

DdPTEN_{G129E}-Halo-TMR exhibits co-localization with PH_{PKB}-eGFP (Fig. 1, 2, 3) because of the lack of substrate (PIP3) binding caused by the G129E mutation. PIP3 induced membrane dissociation of DdPTEN and HsPTEN (Fig. 4c) by inducing a state transition from the stably to primary/weakly binding state (Fig. 5c). However, the membrane dissociation of DdPTEN_{G129E} was induced less effectively than those of DdPTEN and HsPTEN, since the state transition of DdPTEN_{G129E} was not induced by PIP3 (probably because of the impaired substrate binding). Thus, DdPTEN_{G129E} stays bound to the membrane without interacting with PIP3 regardless of PIP3 levels. The above description has been included in the revised manuscript (lines 14-18, pp. 19).

3) Most importantly, the amount of PIP3 in each of these three situations should be quantified and reported before these results are introduced.

Please see our response to comment #1.

4) Result 2 (corresponding to figure 2): It is here that a bistable model should be introduced to make this idea clearer. Why was the magenta line removed as a reference in the middle panel of B? The old position of the line from the top panel in B should be overlaid in the middle panel so as to demonstrate how co-localization occurs and the bistability is lost.

We have submitted another manuscript describing the mathematical model we mention above [<https://www.biorxiv.org/content/early/2018/08/09/356105>]. Thus, the bistable model was not included in this manuscript. However, in the revised manuscript we have included further evidence for the bistability of PIP3 and PTEN (lines 8-20, pp. 10; Fig. 2d, e) and added the magenta lines to the middle panels in Fig. 2b. In addition, please see our response to comment #1 from Reviewer #1 for Fig. 2d and 2e.

5) Result 3 (corresponding to figure 3): In the bottom panel of A, it is claimed that the exclusion of DdPTEN does not occur. Although it does look that way from the figure, this needs quantification for better illustration, and for comparison with the +LY

result as this phenotype seems somewhat weaker. Perhaps something like membrane to cytosol fluorescence intensity ratio is needed. p values are needed in panel D. (p-values are, in general, lacking from all plots in this paper, although the methods claim that the standard t-test was done).

We have added quantification data (Supplementary Fig. 4c,d) and statistics including p values (lines 14-17, pp. 45) in the revised manuscript.

6) Result 4 (corresponding to figure 4): Last paragraph: The scattered plot of the number of molecules between PTEN-Halo-TMR and the membrane. Should there not be a figure reference here?

We have included the scatter plot in Supplementary Fig. 5e.

7) Result 5 (corresponding to figure 5): Error-bars and p-values are missing in plots A and D.

We have included error-bars and p values in Fig. 5a (line 20, pp. 46 – line 2, pp. 47) and 5d (lines 10-12, pp. 47) in the revised manuscript.

Reviewers' Comments:

Reviewer #1:

Remarks to the Author:

The points I raised in my original review have now been addressed to my satisfaction.

Reviewer #3:

Remarks to the Author:

The authors have addressed all of my specific concerns for the paper - and the paper is significantly improved in this regard.

Regarding my overall concern about the notion of bistability for cell motion - the discussion that the authors have added, attempts to address this issue. This discussion is more of a suggestion as to how the chemoattractant receptor pathway is "most likely" directly affecting the bistable PI3K-PTEN system without going through the excitable network. This is possible but needs further proof. However, for the context of this paper I do not think its necessary to go into this proof. I just hope that the title and the discussion tone of the paper suggesting that - "Mutual inhibition generates bistability for cell motility" does not confuse readers - as just the mutual inhibition does not explain the continuous enrichment of PIP3. It brings about bistable states - but does not, in itself, (without the cAMP direct regulation) explain the continuous PIP3 domain.

Regardless, the mutual inhibition mechanism is novel, interesting and well-documented.